# Q-VDiT: Towards Accurate Quantization and Distillation of Video-Generation Diffusion Transformers

**Weilun Feng** [1 2]  **Chuanguang Yang**[✉ 1]  **Haotong Qin** [3]  **Xiangqi Li** [1 2]  **Yu Wang** [1]  **Zhulin An**[✉ 1]  **Libo Huang** [1]
**Boyu Diao** [1]  **Zixiang Zhao** [3]  **Yongjun Xu** [1]  **Michele Magno** [3]

## Abstract

Diffusion transformers (DiT) have demonstrated exceptional performance in video generation. However, their large number of parameters and high computational complexity limit their deployment on edge devices. Quantization can reduce storage requirements and accelerate inference by lowering the bit-width of model parameters. Yet, existing quantization methods for image generation models do not generalize well to video generation tasks. We identify two primary challenges: the loss of information during quantization and the misalignment between optimization objectives and the unique requirements of video generation. To address these challenges, we present **Q-VDiT**, a quantization framework specifically designed for video DiT models. From the quantization perspective, we propose the *Token-aware Quantization Estimator* (TQE), which compensates for quantization errors in both the token and feature dimensions. From the optimization perspective, we introduce *Temporal Maintenance Distillation* (TMD), which preserves the spatiotemporal correlations between frames and enables the optimization of each frame with respect to the overall video context. Our W3A6 Q-VDiT achieves a scene consistency of 23.40, setting a new benchmark and outperforming current state-of-the-art quantization methods by **1.9×**. Code will be available at https://github.com/cantbebetter2/Q-VDiT.

## 1. Introduction

Diffusion models (DMs) (Ho et al., 2020; Song et al., 2020) have demonstrated remarkable success across a wide range

[1]Institute of Computing Technology, Chinese Academy of Sciences [2]University of Chinese Academy of Sciences [3]ETH Zurich. Correspondence to: ✉Zhulin An <anzhulin@ict.ac.cn>, ✉Chuanguang Yang <yangchuanguang@ict.ac.cn>.

*Proceedings of the 42$^{nd}$ International Conference on Machine Learning*, Vancouver, Canada. PMLR 267, 2025. Copyright 2025 by the author(s).

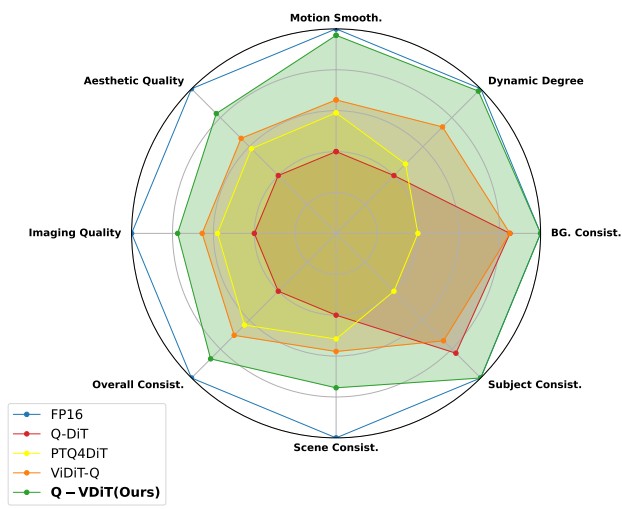

*Figure 1.* Evaluation on VBench of different quantization methods under W3A6 setting.

of generative tasks, including image generation (Ho et al., 2020; Rombach et al., 2022; Dhariwal & Nichol, 2021; Feng et al., 2024; Yang et al., 2025), image super-resolution (Lin et al., 2025; Wang et al., 2024b; Wu et al., 2024b), and video generation (HPC-AI, 2024; Ma et al., 2024). Diffusion Transformers (DiT) (Peebles & Xie, 2023) have emerged as a prominent architecture for generation tasks, leveraging their ability to capture long-range dependencies and scale to large parameter spaces. However, up to billions of parameters (Black-Forest-Labs, 2024) and the increased computational complexity pose significant challenges for their deployment on edge devices.

As an effective model compression technique, quantization reduces the bit-width of parameters and enhances inference speed by utilizing integer operations (Gholami et al., 2022). This approach has been extensively applied to compress CNN-based (Pilipović et al., 2018; Ding et al., 2024) and Transformer-based (Chitty-Venkata et al., 2023) architectures. However, existing quantization methods for Diffusion Transformers (DiT) have primarily focused on image generation tasks, with limited exploration in the context of video

*Figure 2.* Overview of proposed Q-VDiT. The framework includes Token-aware Quantization Estimator (TQE) for forward process and Temporal Maintenance Distillation (TMD) for optimization. The middle part denotes the quantized forward process. $\otimes$ denotes matrix multiplication, $\odot$ denotes token-wise multiplication.

generation. Directly applying existing quantization methods (Li et al., 2023; Ashkboos et al., 2024; Chen et al., 2024; Wu et al., 2024a) to video-generation DiT leads to significant performance degradation (Zhao et al., 2024).

Compared to image generation, video generation models require modeling additional temporal dimensions across multiple frames, significantly increasing the information density relative to image generation tasks. Quantization introduces information loss (Qin et al., 2020; Liu et al., 2020; Feng et al., 2025), which can lead to considerable performance degradation in video generation models, given their higher information density (Zhao et al., 2024). Furthermore, video generation involves strong semantic and temporal correlations between frames. Existing optimization methods (Wu et al., 2024a; He et al., 2023) typically use Mean Squared Error (MSE) to align model outputs directly, which fails to account for these correlations. As a result, these approaches do not calibrate the quantization process from the perspective of the entire video, leading to a degradation in video quality.

To address the aforementioned challenges, we propose Q-VDiT (**Q**uantization of **V**ideo-Generation **Di**ffusion **T**ransformers), a novel framework specifically designed for quantizing video diffusion transformers. Q-VDiT comprises two key components: *Token-aware Quantization Estimator* (TQE) and *Temporal Maintenance Distillation* (TMD). An overview of Q-VDiT is illustrated in Fig. 2. From quantization perspective, quantizing weights introduces significant information loss, which is a primary factor contributing to the degradation of video quality. To mitigate this, we employ a small number of parameters to estimate the low-rank quantization error of the weights from two orthogonal tokens and

feature dimensions. From optimization perspective, relying solely on Mean Squared Error (MSE) for optimization neglects the inter-frame information and fails to capture the overall temporal dynamics of the video. We construct the similarity distribution between frames in the full-precision (FP) model as prior knowledge. Using KL divergence, we align the temporal distributions of the quantized model with the FP model, enabling the quantized model to maintain semantic and temporal coherence across frames. We compare the quantized model's performance in Fig. 1.

The main contributions are summarized as follows:

- We theoretically prove that the quantization error of weights carries less information entropy than the original weights. To address this, we introduce the Token-aware Quantization Estimator (TQE), which employs a small number of additional parameters to perform low-rank approximation of the quantization errors from two orthogonal token and feature dimensions.

- We identify that using MSE alone fails to capture the inter-frame optimization information in video generation. To address this, we propose Temporal Maintenance Distillation (TMD), which models the inter-frame distribution to ensure that the optimization of each frame considers the overall distribution of video characteristics across all frames.

- We propose Q-VDiT, a framework specifically designed for quantizing video diffusion transformers. Extensive experiments on generative benchmarks show that Q-VDiT significantly outperforms current SOTA post-training quantization methods.

## 2. Related Work

### 2.1. Diffusion Model

Diffusion models (Ho et al., 2020; Rombach et al., 2022) perform a forward sampling process by gradually adding noise to the data distribution $\mathbf{x}_0 \sim q(x)$. In DDPM, the forward noise addition process of the diffusion model is a Markov chain, taking the form:

$$q(\mathbf{x}_{1:T}|\mathbf{x}_0) = \prod_{t=1}^{T} q(\mathbf{x}_t|\mathbf{x}_{t-1}),$$

$$q(\mathbf{x}_t|\mathbf{x}_{t-1}) = \mathcal{N}(\mathbf{x}_t; \sqrt{\alpha_t}\mathbf{x}_{t-1}, \beta_t\mathbf{I}), \qquad (1)$$

where $\alpha_t = 1 - \beta_t$, $\beta_t$ is time-related schedule. Diffusion models generate high-quality images by applying a denoising process to randomly sampled Gaussian noise $\mathbf{x}_T \sim \mathcal{N}(\mathbf{0}, \mathbf{I})$, taking the form:

$$p_\theta(\mathbf{x}_{t-1}|\mathbf{x}_t) = \mathcal{N}(\mathbf{x}_{t-1}; \hat{\mu}_{\theta,t}(\mathbf{x_t}), \hat{\beta}_t\mathbf{I}), \qquad (2)$$

where $\hat{\mu}_{\theta,t}$ and $\hat{\beta}_t$ are outputed by the diffusion model.

### 2.2. Diffusion Quantization

For diffusion model quantization, methods such as Q-DM (Li et al., 2024b), BinaryDM (Zheng et al., 2024b), BiDM (Zheng et al., 2024a), and TerDiT (Lu et al., 2024) use quantization-aware training to maintain model performance under 1-2 bits. However, these approaches require extensive additional training time, often lasting several days. For more efficient quantization, approaches like Q-Diffusion (Li et al., 2023), PTQ4DM (Shang et al., 2023), PTQ-D (He et al., 2024), TFMQ-DM (Huang et al., 2024a), QuEST (Wang et al., 2024a), EfficientDM (He et al., 2023), and MixDQ (Zhao et al., 2025) explore quantization from the perspectives of quantization error, temporal features, and calibration data, particularly for Unet-based diffusion models. Similarly, Q-DiT (Chen et al., 2024), PTQ4DiT (Wu et al., 2024a), SVDQuant (Li et al., 2024a), and ViDiT-Q (Zhao et al., 2024) focus on the quantization of diffusion transformers, considering their unique data distributions and computational characteristics. However, existing quantization methods primarily focus on image generation tasks, with limited exploration into the more challenging domain of video generation. Therefore, this paper focuses on optimizing the quantization performance of video-generation diffusion transformers.

## 3. Methods

### 3.1. Model Quantization

Model quantization maps model weights and activations to low bit integer values to reduce memory footprint and

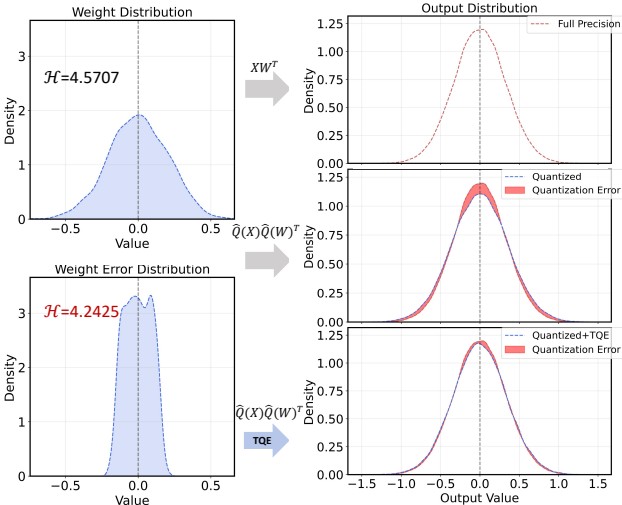

*Figure 3.* An illustration of TQE in Q-VDiT

accelerate the inference. For a floating vector $\mathbf{x}_f$, the quantization process can be formulated as

$$\hat{\mathbf{x}}_q = Q(\mathbf{x}_f, s, z) = clip(\lfloor\frac{\mathbf{x}_f}{s}\rceil + z, 0, 2^N - 1),$$

$$s = \frac{u - l}{2^N - 1}, z = -\lfloor\frac{l}{s}\rceil, \qquad (3)$$

where $\hat{\mathbf{x}}_q$ indicates quantized vector in integer, $\lfloor\cdot\rceil$ is round fuction and $clip(\cdot)$ is function that clamps values into the range of $[0, 2^N-1]$, $s$ is a scale factor and $z$ is a quantization zero point. $l$ and $u$ are the lower and upper bounds of quantization thresholds, respectively. They are determined by $\mathbf{x}_f$ and the target bit-width. Reversely, in order to restore the low-bit integer quantization vector $\hat{\mathbf{x}}_q$ to the full precision representation, the dequantization process is formulated as

$$\hat{\mathbf{x}}_f = \hat{Q}(\mathbf{x}_f) = (\hat{\mathbf{x}}_q - z)s, \qquad (4)$$

where $\hat{\mathbf{x}}_f$ is the dequantized vector used for forward process.

### 3.2. Token-aware Quantization Estimator

For Diffusion Transformers (DiTs) (Peebles & Xie, 2023; Ma et al., 2024), the latent representation of the generation target is denoted as $\mathbf{Z} \in \mathbb{R}^{n \times d}$, where $n$ is the number of tokens and $d$ is the hidden dimension. In image generation (Peebles & Xie, 2023), $n$ corresponds to the spatial token count $s$. However, for video generation (Ma et al., 2024), $n = s \times t$, where $s$ is the spatial token number and $t$ is the temporal token number, representing $t$ frames. This means that video DiTs (V-DiTs) contain significantly more tokens than image DiTs (I-DiTs), greatly enhancing their expressive capacity. However, quantization, particularly low-bit quantization, can result in substantial information loss, which is especially critical for V-DiTs (Zhao et al., 2024).

Therefore, our goal is to retain as much of the model's information as possible using a minimal number of parameters, thereby preserving the expressive capability of V-DiTs.

**Proposition 3.1.** *Given a L layer model* $f\{\mathbf{W}_{i=1}^L\}$, *the quantization process for weight is equivalent to applying a perturbation* $\Delta$ *to the original weight:*

$$f\{\sum_{i=1}^L \hat{Q}(\mathbf{W}_i)\} = f\{\sum_{i=1}^L \mathbf{W}_i + \Delta_i\}, \quad (5)$$

*where* $\mathbf{W}_i$ *stands for i-th layer weight.*

Therefore, our goal is to express the quantization error $\Delta$ in terms of a set of efficient parameters.

**Theorem 3.2.** *Given any layer weight* $\mathbf{W}_i$, *the corresponding quantization error* $\Delta_i$ *has less information entropy compared with original weight* $\mathbf{W}_i$:

$$\mathcal{H}(\Delta_i) \leq \mathcal{H}(\mathbf{W}_i), \quad (6)$$

*where* $\mathcal{H}(\cdot)$ *denotes the information entropy calculation.*

Theorem 3.2 indicates that although the dimension of $\Delta_i$ remains unchanged, its information entropy is lower than that of the original weight $\mathbf{W}i$. Therefore, compared to the original weight dimensions, the quantization error can be estimated in a lower-rank space and represented using fewer parameters. To achieve this, we use two low-dimensional vector parameters, $\alpha$ and $\beta$, to represent the quantization errors through low-rank estimation. For matrix multiplication with weight $\mathbf{W} \in \mathbb{R}^{d_{out} \times d_{in}}$ and input $\mathbf{X} \in \mathbb{R}^{n \times d_{in}}$, we approximate the operation during quantization as follows:

$$\dot{\Delta} := \mathbf{X}\alpha,$$
$$\mathbf{X}\mathbf{W}^\top \approx \mathbf{X}\hat{Q}(\mathbf{W})^\top + \dot{\Delta}\beta, \quad (7)$$

where $\alpha \in \mathbb{R}^{d_{in}}$ and $\beta \in \mathbb{R}^{d_{out}}$. We define $\dot{\Delta} \in \mathbb{R}^{n \times 1}$ to represent the low-rank estimated quantization error, with $\beta$ aligning the output dimension. By employing this approach, the additional parameters required to approximate the quantization error are reduced from the original weight size of $(d_{out} \times d_{in})$ to only $(d_{out} + d_{in})$, significantly lowering the parameter overhead. Furthermore, the computational efficiency of the model is preserved after quantization.

But for latent representation of the video generation target, $\mathbf{Z} \in \mathbb{R}^{n \times d}$, the information is distributed across both the external token dimension and the internal feature dimension. When activations are quantized, the degree of information loss varies across tokens (He et al., 2023; Ashkboos et al., 2024). Consequently, error estimation based solely on the feature dimension fails to account for the variation in quantization information loss among tokens, leading to suboptimal quantization performance.

To address this issue, we propose Token-aware Quantization Estimator (TQE) to estimate overall quantization error across both the token and feature dimensions. The illustration of TQE is in Fig. 3. In the token dimension, as information at different token positions tends to be highly concentrated (Zhao et al., 2024), we pre-scale the quantized activations to estimate the quantization information loss effectively. Additionally, given the unique temporal differences across frames in video generation (Ma et al., 2024; HPC-AI, 2024), we selectively scale the temporal tokens of individual frames to ensure balanced information distribution across frames. Formally, we reformulate Eq. (7) as:

$$\hat{\Delta}_{[f_i+1:f_i+s,:]} := (\mathcal{M}_i \bigodot \hat{Q}(\mathbf{X})_{[f_i+1:f_i+s,:]})\alpha, \ \mathcal{M} \in \mathbb{R}^t$$
$$\mathbf{X}\mathbf{W}^\top \approx \hat{Q}(\mathbf{X})\hat{Q}(\mathbf{W})^\top + \hat{\Delta}\beta, \quad (8)$$

where $\mathbf{X} \in \mathbb{R}^{n \times d_{in}}$, $n = s \times t$, $s$ represents the spatial token number, and $t$ denotes the temporal token number. We define $f_i := i \times s$, where $i \in [1, 2, \cdots, t]$. To enable $\mathcal{M}$ to account for the activation quantization error, we jointly consider the weight of the token's salient measurement and the dissimilarity introduced by activation quantization when initializing $\mathcal{M}$. The initialization of $\mathcal{M}$ is defined as:

$$\eta_i = \frac{\exp[1 - \rho(\mathbf{X}_{[f_i+1:f_i+s,:]}, \hat{Q}(\mathbf{X})_{[f_i+1:f_i+s,:]})]}{\sum_{\nu=0}^{t-1} \exp[1 - \rho(\mathbf{X}_{[f_\nu+1:f_\nu+s,:]}, \hat{Q}(\mathbf{X})_{[f_\nu+1:f_\nu+s,:]})]},$$
$$\omega_i = \frac{\sum_{\tau=f_i+1}^{f_i+s} |\mathbf{X}_{[\tau,:]}|}{\sum_{\nu=0}^{t-1}[\sum_{\tau=f_\nu+1}^{f_\nu+s} |\mathbf{X}_{[\tau,:]}|]},$$
$$\mathcal{M}_i = \frac{\eta_i}{\omega_i}, \quad (9)$$

where $\rho(\cdot, \cdot)$ computes the similarity between two sequences, and $\eta_i$ serves as the weighting factor for the activation quantization error of frame $t$. To ensure value balance, the weighting factor is normalized by the salient measurement $\omega_i$ of the token sequences in frame $t$.

Overall, we initialize $\mathcal{M}$ using Eq. (9), set $\alpha$ with Kaiming initialization (He et al., 2015), and initialize $\beta$ to zero following common practices (Hu et al., 2021; He et al., 2023). The modified quantization forward process is then implemented using Eq. 8, replacing the original forward computation. Leveraging the LoRunner Kernel (Li et al., 2024a), this replacement process introduces negligible latency. Details can be found in the Appendix Sec. F.

### 3.3. Temporal Maintenance Distillation

For diffusion transformer optimization, existing methods (Wu et al., 2024a; He et al., 2023) optimize the training parameters of the quantization model by minimizing the

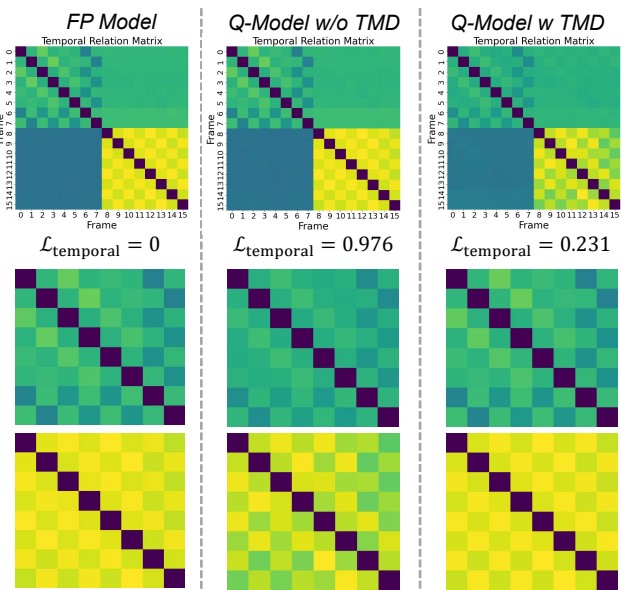

Figure 4. An illustration of TMD in Q-VDiT. We have enlarged the upper left and lower right corners additionally.

output error between the full-precision (FP) model and the quantization model by

$$\mathcal{L}_{task} = ||O_{FP}(\mathbf{X}; \mathbf{W}), O_Q(\hat{Q}(\mathbf{X}); \hat{Q}(\mathbf{W}))||^2, \quad (10)$$

where $O_{FP}(\mathbf{X}; \mathbf{W}) \in \mathbb{R}^{n \times d}$ represents the output of the full-precision model, which serves as the ground truth, while $O_Q(\hat{Q}(\mathbf{X}); \hat{Q}(\mathbf{W})) \in \mathbb{R}^{n \times d}$ denotes the quantized output. To simplify the notation, we redefine the outputs of both models as $\mathbf{S}^{FP}$ and $\mathbf{S}^Q$, where $\mathbf{S}^{FP}$ and $\mathbf{S}^Q$ represent the restored image information required for each frame in the generated video. Accordingly, Eq. (10) is rewritten as:

$$\begin{aligned} \mathcal{L}_{task} &= ||\mathbf{S}^{FP}, \mathbf{S}^Q||^2 \\ &= \sum_{i=0}^{t-1} ||\mathbf{S}^{FP}_{[f_i+1:f_i+s,:]}, \mathbf{S}^Q_{[f_i+1:f_i+s,:]}||^2, \end{aligned} \quad (11)$$

In contrast to image generation which involves the image information of a single frame, video generation requires $\mathbf{S}$ to capture the image information across multiple frames. For the quantization model that needs to be optimized, the gradient for each frame $i$ in $\mathbf{S}^Q$, derived from Eq. (11) is given by:

$$\frac{\partial \mathcal{L}_{task}}{\partial \mathbf{S}^Q_{[f_i+1:f_i+s,:]}} = -2(\mathbf{S}^{FP}_{[f_i+1:f_i+s,:]} - \mathbf{S}^Q_{[f_i+1:f_i+s,:]}), \quad (12)$$

In this approach, the optimization of each frame in the quantized model only considers the information gap between it and the corresponding frame in the FP model. However, the information between different frames is not independent in a complete video (Huang et al., 2024b; Ma et al., 2024). Therefore, using Eq. (11) alone to optimize a video generation quantized model is insufficient. It neglects the inter-frame information dependencies, which may lead to sub-optimal optimization results (Yang et al., 2022) and fail to ensure the coherence of video content.

To directly perceive the inter-frame information during optimization, we introduce Temporal Maintenance Distillation as illustrated in Fig. 4. We calculate the relationship distribution between token sequences from different frames, allowing for direct interaction of information between frames. We use KL divergence to minimize the gap between them. Formally, we compute the temporal relation between the $i$-th and $j$-th frames as follows:

$$\begin{aligned} \mathbf{T}_{i,j} &= \rho(\mathbf{S}_{[f_i+1:f_i+s,:]}, \mathbf{S}_{[f_j+1:f_j+s,:]}), \\ & i, j \in [1, 2, \cdots, t], \end{aligned} \quad (13)$$

where $\mathbf{T}_{i,j}$ presents the relationship between $i$-th frame and $j$-th frame. Thus, we can formulate the temporal relation distribution for the $i$-th frame by

$$\mathbf{D}_i = \mathrm{softmax}(\mathrm{concat}[\mathbf{T}_{i,1}, \cdots, \mathbf{T}_{i,t}]), \mathbf{D}_i \in \mathbb{R}^t, \quad (14)$$

where $\mathbf{D}_i$ presents the temporal relation distribution between $i$-th frame and whole video frames, and we use softmax to formulate probability distribution. For $\mathbf{S}^{FP}$ and $\mathbf{S}^Q$ from the full-precision model and quantized model, we can formulate $\mathbf{D}^{FP}$ and $\mathbf{D}^Q$ correspondingly. Thus, we present Temporal Maintenance Distillation by

$$\mathcal{L}_{temporal} = \sum_{i=1}^{t} \mathrm{KL}(\mathbf{D}_i^{FP}, \mathbf{D}_i^Q), \quad (15)$$

From Eq. (15), we can derive the gradient for single frame $\mathbf{S}^Q_{[f_i+1:f_i+s,:]}$ by

$$\begin{aligned} \frac{\partial \mathcal{L}_{temporal}}{\partial \mathbf{S}^Q_{[f_i+1:f_i+s,:]}} = \sum_{j=1}^{t} &\left[ \frac{\partial \mathcal{L}_{temporal}}{\partial \mathbf{T}^Q_{i,j}} \cdot \frac{\partial \mathbf{T}^Q_{i,j}}{\partial \mathbf{S}^Q_{[f_i+1:f_i+s,:]}} + \right. \\ &\left. \frac{\partial \mathcal{L}_{temporal}}{\partial \mathbf{T}^Q_{j,i}} \cdot \frac{\partial \mathbf{T}^Q_{j,i}}{\partial \mathbf{S}^Q_{[f_i+1:f_i+s,:]}} \right], \end{aligned} \quad (16)$$

*Table 1.* Performance of text-to-video generation on VBench evaluation benchmark suite. The bit-width "16" represents FP16 without quantization. **Bold**: the best result. Underline: the second-best result.

| Method | Bit-width (W/A) | Imaging Quality | Aesthetic Quality | Motion Smooth. | Dynamic Degree | BG. Consist. | Subject Consist. | Scene Consist. | Overall Consist. |
|---|---|---|---|---|---|---|---|---|---|
| - | 16/16 | 63.68 | 57.12 | 96.28 | 56.94 | 96.13 | 90.28 | 39.61 | 26.21 |
| Q-DiT | 4/6 | 55.99 | 52.99 | 95.94 | 51.39 | 94.85 | 86.40 | 32.67 | 24.50 |
| PTQ4DiT | 4/6 | 55.58 | 53.31 | 94.57 | 56.94 | 93.70 | 86.90 | 33.65 | 23.48 |
| SmoothQuant | 4/6 | 54.16 | 52.20 | 94.83 | 55.56 | 93.55 | 87.08 | 31.40 | 22.54 |
| Quarot | 4/6 | 53.79 | 51.95 | 93.26 | 51.39 | 93.15 | 85.26 | 32.77 | 22.89 |
| EfficientDM | 4/6 | 55.63 | 53.92 | 95.19 | 51.39 | 95.10 | 86.58 | 35.90 | 23.90 |
| SVDQuant | 4/6 | 55.23 | 53.38 | 95.90 | 47.22 | 95.70 | 87.76 | 32.19 | 23.18 |
| ViDiT-Q | 4/6 | 55.63 | 53.68 | 96.13 | 56.94 | 95.38 | 86.94 | 32.70 | 24.53 |
| **Q-VDiT** | 4/6 | **57.49** | **55.18** | **96.25** | **68.06** | **95.72** | **87.78** | **38.66** | **25.02** |
| Q-DiT | 3/8 | 36.23 | 31.97 | 86.77 | 13.89 | 95.86 | 90.09 | 0.08 | 10.29 |
| PTQ4DiT | 3/8 | 46.98 | 40.88 | 93.63 | 16.67 | 92.38 | 84.57 | 7.63 | 18.04 |
| SmoothQuant | 3/8 | 36.86 | 33.24 | 91.32 | 27.78 | 93.16 | 82.42 | 1.02 | 11.03 |
| Quarot | 3/8 | 35.94 | 32.99 | 90.76 | 27.78 | 93.65 | 83.44 | 2.51 | 11.14 |
| EfficientDM | 3/8 | 49.97 | 43.78 | 95.49 | 31.94 | 95.92 | 89.14 | 13.99 | 16.61 |
| SVDQuant | 3/8 | 43.54 | 32.51 | 90.52 | 23.61 | 94.28 | 83.68 | 4.65 | 10.43 |
| ViDiT-Q | 3/8 | 50.62 | 42.83 | 95.84 | 31.94 | **96.35** | 88.06 | 15.19 | 18.53 |
| **Q-VDiT** | 3/8 | **54.94** | **49.94** | **97.11** | **50.00** | 95.96 | **90.14** | **25.07** | **22.39** |
| Q-DiT | 3/6 | 37.47 | 32.58 | 86.02 | 13.89 | 94.84 | 88.72 | 0.07 | 10.57 |
| PTQ4DiT | 3/6 | 45.31 | 40.20 | 93.43 | 19.45 | 91.59 | 83.15 | 7.85 | 16.55 |
| SmoothQuant | 3/6 | 36.88 | 33.57 | 91.18 | 33.33 | 93.02 | 81.99 | 1.24 | 11.01 |
| Quarot | 3/6 | 36.02 | 32.54 | 91.15 | 23.61 | 93.86 | 82.14 | 2.26 | 11.06 |
| EfficientDM | 3/6 | 48.82 | 42.11 | 95.04 | 34.72 | 95.17 | 88.17 | 12.04 | 15.85 |
| SVDQuant | 3/6 | 43.67 | 32.32 | 90.28 | 23.61 | 94.18 | 83.09 | 3.56 | 10.30 |
| ViDiT-Q | 3/6 | 48.76 | 42.70 | 95.51 | 37.50 | 95.34 | 86.86 | 11.99 | 18.38 |
| **Q-VDiT** | 3/6 | **53.60** | **49.66** | **96.98** | **55.56** | **95.41** | **89.06** | **23.40** | **22.58** |

For any $\frac{\partial \mathcal{L}_{temporal}}{\partial \mathbf{T}_{i,j}^Q}$, we can calculate it using the chain rule

$$
\begin{aligned}
\frac{\partial \mathcal{L}_{temporal}}{\partial \mathbf{T}_{i,j}^Q} &= \sum_{k=1}^{t} \frac{\partial \mathcal{L}_{temporal}}{\partial \mathbf{D}_{i,k}^Q} \cdot \frac{\partial \mathbf{D}_{i,k}^Q}{\partial \mathbf{T}_{i,j}^Q} \\
&= -\mathbf{D}_{i,j}^{FP}(1 - \mathbf{D}_{i,j}^Q) + \sum_{k \neq j} \mathbf{D}_{i,k}^{FP} \mathbf{D}_{i,j}^Q \quad (17) \\
&= \sum_{k} \mathbf{D}_{i,k}^{FP} \mathbf{D}_{i,j}^Q - \mathbf{D}_{i,j}^{FP},
\end{aligned}
$$

From Eq. (17), any correlation between $i$-th and $j$-th frame $\mathbf{T}_{i,j}^Q$ is numerically affected by all frames which ensures the perception of overall video information. For any sequences $\mathbf{v}_i$ and $\mathbf{v}_j$ stands for $i$-th and $j$-th frame used in Eq. (13), the gradient for $i$-th frame $\frac{\partial \mathbf{T}_{i,j}}{\partial \mathbf{v}_i}$ can be derived by

$$
\begin{aligned}
\frac{\partial \mathbf{T}_{i,j}^Q}{\partial \mathbf{v}_i} &= \frac{\|\mathbf{v}_i\|\|\mathbf{v}_j\|\mathbf{v}_j - (\mathbf{v}_i \cdot \mathbf{v}_j)\frac{\mathbf{v}_i}{\|\mathbf{v}_i\|}}{\|\mathbf{v}_i\|^2 \|\mathbf{v}_j\|^2}, \\
\mathbf{v}_i &:= \mathbf{S}_{[f_i+1:f_i+s,:]}^Q,
\end{aligned} \quad (18)
$$

By applying Eq. (17) and Eq. (18) to Eq. (16), the optimization of any single frame $\mathbf{S}_{[f_i+1:f_i+s,:]}^Q$ is jointly guided by all

frames. This enables direct interaction between frames and optimizes the temporal coherence across the entire video. $\mathcal{L}_{temporal}$ effectively compensates for the lack of alignment that arises when focusing solely on single-frame optimizations. Through this approach, the video information from both the FP network and the quantized network is enhanced by capturing the overall temporal representations of the video. In summary, the overall optimization objective can be reformulated as follows:

$$
\mathcal{L}_{total} = \mathcal{L}_{task} + \gamma \mathcal{L}_{temporal} \quad (19)
$$

## 4. Experiments

### 4.1. Experimental and Evaluation Settings

**Experimental Settings:** Following previous work ViDiT-Q (Zhao et al., 2024), we apply our Q-VDiT to Open-SORA (HPC-AI, 2024) and Latte (Ma et al., 2024) for video generation task. We mainly focus on harder settings of W4A6 (4-bit weight quantization and 6-bit activation quantization), W3A8, and W3A6. We follow the exper-

*Table 2.* Performance of text-to-video generation on OpenSORA prompt set. **Bold**: the best result. Underline: the second-best result.

| Method | Bit-width (W/A) | CLIPSIM | CLIP-Temp | VQA-Aesthetic | VQA-Technical | Δ FLOW Score.(↓) | Warping Error (↓) |
|---|---|---|---|---|---|---|---|
| - | 16/16 | 0.1797 | 0.9986 | 66.91 | 53.49 | - | 0.016 |
| Q-DiT | 4/6 | 0.1757 | 0.9987 | 57.11 | 38.12 | 0.913 | 0.015 |
| PTQ4DiT | 4/6 | 0.1777 | 0.9978 | 62.37 | 37.68 | 0.287 | 0.022 |
| SmoothQuant | 4/6 | 0.1781 | 0.9981 | 55.66 | 22.37 | 1.076 | 0.019 |
| Quarot | 4/6 | 0.1778 | 0.9980 | 56.24 | 30.75 | 1.013 | 0.018 |
| EfficientDM | 4/6 | 0.1753 | 0.9986 | 62.43 | 43.85 | 0.649 | 0.021 |
| SVDQuant | 4/6 | 0.1780 | 0.9983 | 64.84 | 36.24 | 0.956 | 0.015 |
| ViDiT-Q | 4/6 | 0.1782 | 0.9985 | 54.66 | 49.78 | 0.306 | **0.012** |
| **Q-VDiT** | 4/6 | **0.1784** | **0.9989** | **67.05** | **53.75** | **0.281** | 0.013 |
| Q-DiT | 3/8 | 0.1707 | 0.9955 | 22.31 | 3.84 | 1.167 | 0.018 |
| PTQ4DiT | 3/8 | 0.1764 | 0.9952 | 35.21 | 8.56 | 1.389 | 0.039 |
| SmoothQuant | 3/8 | 0.1758 | 0.9965 | 16.55 | 1.08 | 1.227 | 0.064 |
| Quarot | 3/8 | 0.1755 | 0.9961 | 26.59 | 5.66 | 1.048 | 0.038 |
| EfficientDM | 3/8 | 0.1762 | 0.9987 | 45.67 | 28.42 | 1.306 | 0.017 |
| SVDQuant | 3/8 | 0.1735 | 0.9986 | 42.13 | 18.96 | 1.457 | 0.019 |
| ViDiT-Q | 3/8 | 0.1733 | 0.9985 | 41.00 | 13.48 | 1.619 | 0.020 |
| **Q-VDiT** | 3/8 | **0.1766** | **0.9988** | **54.92** | **61.59** | **0.839** | **0.011** |
| Q-DiT | 3/6 | 0.1681 | 0.9961 | 11.90 | 0.69 | 1.412 | 0.028 |
| PTQ4DiT | 3/6 | 0.1765 | 0.9947 | 26.35 | 5.42 | 1.185 | 0.042 |
| SmoothQuant | 3/6 | 0.1768 | 0.9959 | 18.07 | 1.33 | 1.286 | 0.068 |
| Quarot | 3/6 | 0.1762 | 0.9950 | 25.89 | 3.95 | 1.084 | 0.042 |
| EfficientDM | 3/6 | 0.1747 | 0.9982 | 43.54 | 29.58 | 1.217 | 0.020 |
| SVDQuant | 3/6 | 0.1712 | 0.9975 | 40.75 | 16.51 | 1.656 | 0.026 |
| ViDiT-Q | 3/6 | 0.1716 | 0.9984 | 39.82 | 10.26 | 1.695 | 0.020 |
| **Q-VDiT** | 3/6 | **0.1785** | **0.9986** | **53.53** | **59.10** | 0.914 | **0.012** |

imental settings in ViDiT-Q (Zhao et al., 2024). For the Open-Sora (HPC-AI, 2024) model, we use the setting in Appendix Sec. D.1 for benchmark evaluation and use 10 prompts provided by OpenSora prompt sets to generate 10 videos for multi-aspects metrics evaluation. More experiments and details can be found in the Sec. 4.3, Appendix Sec. B, and E.

**Evaluation Settings:** The evaluation contains two settings like ViDiT-Q (Zhao et al., 2024). We first evaluate the quantized model on VBench (Huang et al., 2024b) to provide comprehensive results for **benchmark evaluation**. To align with ViDiT-Q, we select 8 major dimensions from Vbench. We select representative metrics and measure them on Open-SORA prompt sets for **multi-aspects metrics evaluation** like ViDiT-Q. Following EvalCrafter (Liu et al., 2024), we select CLIPSIM, CLIP-Temp, Warping Error to measure consistency, and DOVER (Wu et al., 2023) video quality assessment (VQA) metrics to evaluate the generation quality, Flow-score are used for evaluating the temporal consistency. For Open-SORA model, we use 100-step DDIM with CFG scale of 4.0. For Latte, we adopt the class-conditioned Latte model trained on UCF-101 and use the 20-step DDIM solver with CFG scale of 7.0. More details can be found in

Appendix Sec. D.

**Compared Methods:** We compare different baseline PTQ methods. For LLM baseline, we compare SmoothQuant (Xiao et al., 2023) and Quarot (Ashkboos et al., 2024). For DM baseline, we mainly compare EfficientDM (He et al., 2023) and SVDQuant (Li et al., 2024a). For DiT baseline, we mainly compare Q-DiT (Chen et al., 2024), PTQ4DiT (Wu et al., 2024a), and ViDiT-Q (Zhao et al., 2024). It is worth mentioning that only ViDiT-Q and Q-DiT have conducted video-generation task evaluation. We rerun all compared methods for fair comparison.

### 4.2. Main Results

**Benchmark evaluation:** In benchmark evaluation, we use vbench (Huang et al., 2024b) to evaluate quantization video diffusion transformer from three key video aspects among 8 details dimensions. As present in Tab. 1, our proposed Q-VDiT achieves notable improvement across dimensions under W4A6, W3A8, and W3A6 settings compared with all existing quantization methods. Especially under the hardest 3-bit weight settings, Q-VDiT exceeds the optimal results of existing methods by a large margin. Like in W3A6 setting,

*Table 3.* Higher bit setting performance of text-to-video generation on OpenSORA prompt set.

| Method | Bit-width (W/A) | CLIPSIM | CLIP-Temp | VQA-Aesthetic | VQA-Technical | Δ FLOW Score.(↓) | Warping Error (↓) |
|--------|-----------------|---------|-----------|---------------|---------------|------------------|-------------------|
| - | 16/16 | 0.1797 | 0.9986 | 66.91 | 53.49 | - | 0.016 |
| Q-Diffusion | 8/8 | 0.1781 | 0.9987 | 51.68 | 38.27 | 0.328 | 0.024 |
| Q-DiT | 8/8 | 0.1788 | 0.9977 | 61.03 | 34.97 | 0.473 | 0.017 |
| PTQ4DiT | 8/8 | 0.1836 | 0.9991 | 54.56 | 53.33 | 0.440 | 0.018 |
| SmoothQuant | 8/8 | 0.1951 | 0.9986 | 59.78 | 51.53 | 0.331 | 0.019 |
| Quarot | 8/8 | 0.1949 | 0.9976 | 58.73 | 52.28 | 0.215 | 0.020 |
| ViDiT-Q | 8/8 | 0.1950 | 0.9991 | 60.70 | 54.64 | 0.089 | 0.016 |
| **Q-VDiT** | 8/8 | 0.1950 | 0.9987 | 64.43 | 56.40 | 0.099 | 0.015 |
| Q-DiT | 6/6 | 0.1710 | 0.9943 | 11.04 | 1.869 | 41.10 | 0.024 |
| PTQ4DiT | 6/6 | 0.1799 | 0.9976 | 59.97 | 43.89 | 0.997 | 0.019 |
| SmoothQuant | 6/6 | 0.1807 | 0.9985 | 56.45 | 48.21 | 29.26 | 0.020 |
| Quarot | 6/6 | 0.1820 | 0.9975 | 61.47 | 53.06 | 0.146 | 0.023 |
| ViDiT-Q | 6/6 | 0.1791 | 0.9984 | 64.45 | 51.58 | 0.625 | 0.016 |
| **Q-VDiT** | 6/6 | 0.1798 | 0.9984 | 67.31 | 54.03 | 0.061 | 0.019 |
| Q-DiT | 4/8 | 0.1687 | 0.9833 | 0.007 | 0.018 | 3.013 | 0.018 |
| PTQ4DiT | 4/8 | 0.1735 | 0.9973 | 2.210 | 0.318 | 0.108 | 0.024 |
| SmoothQuant | 4/8 | 0.1832 | 0.9983 | 31.96 | 22.85 | 0.415 | 0.021 |
| Quarot | 4/8 | 0.1817 | 0.9965 | 47.36 | 33.13 | 0.326 | 0.020 |
| ViDiT-Q | 4/8 | 0.1809 | 0.9989 | 60.62 | 49.38 | 0.153 | 0.012 |
| **Q-VDiT** | 4/8 | 0.1811 | 0.9989 | 71.32 | 55.56 | 0.147 | 0.012 |

*Table 4.* Ablation studies of Q-VDiT techniques on W3A8 Open-Sora model.

| Method | Bit-width (W/A) | Imaging Quality | Aesthetic Quality | Motion Smooth. | Dynamic Degree | BG Consist. | Subject Consist. | Scene Consist. | Overall Consist. |
|--------|-----------------|-----------------|-------------------|----------------|----------------|-------------|------------------|----------------|------------------|
| - | 16/16 | 63.68 | 57.12 | 96.28 | 56.94 | 96.13 | 90.28 | 39.61 | 26.21 |
| PTQ4DiT | 3/6 | 45.31 | 40.20 | 93.43 | 19.45 | 91.59 | 83.15 | 7.85 | 16.55 |
| +TQE (w/o $\mathcal{M}$) | 3/6 | 51.36 | 47.69 | 96.14 | 47.22 | 94.86 | 88.21 | 21.94 | 21.65 |
| +TQE (w $\mathcal{M}$) | 3/6 | 52.35 | 48.97 | 96.77 | 51.39 | 95.12 | 88.86 | 22.38 | 22.00 |
| +TMD | 3/6 | 53.46 | 48.70 | 96.00 | 52.78 | 95.24 | 89.01 | 22.87 | 22.31 |
| **Q-VDiT** | 3/6 | **53.60** | **49.66** | **96.98** | **55.56** | **95.41** | **89.06** | **23.40** | **22.58** |

Q-VDiT improves SOTA scene consistency from 12.04 to 23.40 which is almost doubled.

**Multi-aspects metrics evaluation:** In multi-aspects metrics evaluation, Q-VDiT also achieves notable improvement across all metrics in Tab. 2. Q-VDiT achieves almost lossless performance under W4A6 setting and greatly improves existing quantization methods under the hardest 3-bit weight settings. In W3A6 setting, Q-VDiT improve VQA-Technical from current SOTA 29.58 to 59.10, CLIPSIM from 0.1768 to 0.1785 by a great margin.

### 4.3. Additional Results on Higher Bit Settings

Tab. 1 shows that Q-VDiT achieves notable improvement under relatively harder settings with 3-4bit weight quantization. In Tab. 3, we present more bit settings on Open-Sora (HPC-AI, 2024) model quantization. It can be seen that our Q-VDiT still outperforms the current quantization

*Table 5.* Efficiency comparisons of different methods across GPU memory and time under W8A8 setting.

| Method | GPU Memory (MB) | GPU Time (hours) | VQA-Aesthetic |
|--------|-----------------|------------------|---------------|
| ViDiT-Q | 16600 | 12.5 | 60.70 |
| EfficientDM | 19460 | 12.6 | 61.25 |
| PTQ4DiT | 17650 | 12.8 | 54.56 |
| **Q-VDiT** | 18770 | 12.9 | **64.43** |

methods under higher bit settings. On W4A8 setting, Q-VDiT achieves 71.32 VQA- Aesthetic which is even higher than the FP model. This further proves our Q-VDiT's superiority at different bits. Notably, Q-VDiT achieves the best VQA metrics under W4A8 setting which even surpass the full-precision performance.

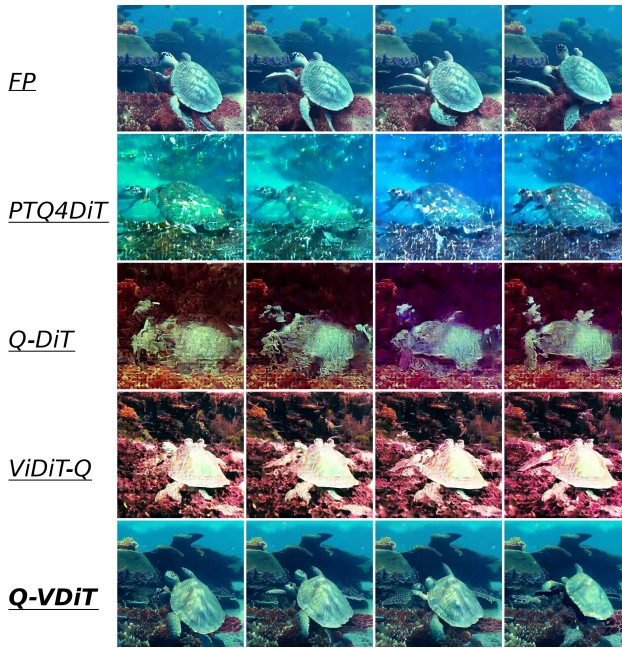

Figure 5. Visualization of different frames in a single video.

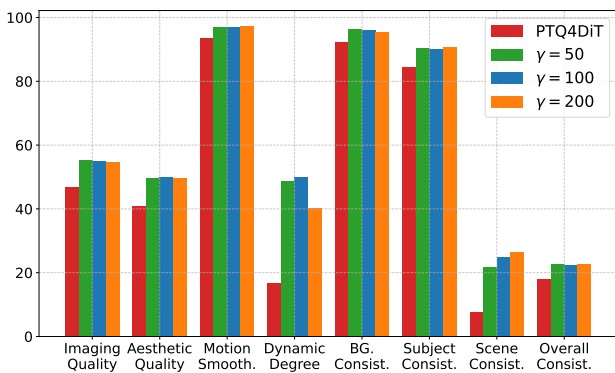

Figure 6. Ablation study on different $\gamma$ in TMD.

## 4.4. Qualitative Comparison

In Fig. 5, we present different frames in a single video under W3A6 with the same prompt to qualitatively compare video generation ability. We evenly sample 4 frames from the complete video. All other comparison methods may not even produce clear images, and some may not even produce meaningful images. Our proposed Q-VDiT not only produces clear and meaningful images but also has significant coherent motion changes between different frames. This indicates that Q-VDiT can still produce good and meaningful complete videos even when other methods fail. More results can be found in Appendix Sec. G.

## 4.5. Ablation Study

**Different techniques of Q-VDiT:** In Tab. 4, we compared different proposed techniques used in Q-VDiT. All proposed techniques can improve the performance of quantized model. We also compare TQE without $\mathcal{M}$. After introducing token-aware $\mathcal{M}$, the effect of TQE is further enhanced which proves the necessity of considering both token and feature dimensions. By combining TQE and TMD, our Q-VDiT achieves the best results across all 8 dimensions.

**Different hyperparameters used in TMD:** In Fig. 6, we compare different $\gamma$ used in Eq. (19) and PTQ baseline method PTQ4DiT (Wu et al., 2024a). We conduct experiment on W3A8 Open-Sora model. It can be seen that all different $\gamma$ used in TMD can notably enhance model perfor-

mance compared with baseline PTQ4DiT. This shows that our TMD is not sensitive to hyperparameter selection. In our practice, we use $\gamma = 100$ for balanced choice.

## 4.6. Training Resource Cost

In Tab. 5, we present the training cost of GPU memory cost and time cost between different PTQ methods under W8A8 setting. Compared to no-calibration method ViDiT-Q, we only bring 3% additional time cost and less compared with other calibration methods. Yet we achieve the best video quality metrics.

## 5. Conclusion

In this paper, we have proposed Q-VDiT, a quantization method tailored specifically for video Diffusion Transformers. To address severe model quantization information loss, we have proposed Token-aware Quantization Estimator to compensate for quantization errors from both token and feature dimensions. To maintain the spatiotemporal correlation between different frames in videos, we have proposed Temporal Maintenance Distillation to optimize each frame from the perspective of the overall video. Our extensive experiments have demonstrated the superiority of Q-VDiT over baseline and other previous quantization methods.

**Acknowledgements.** This work is partially supported by the National Natural Science Foundation of China under Grant Number 62476264 and 62406312, the Postdoctoral Fellowship Program and China Postdoctoral Science Foundation under Grant Number BX20240385 (China National Postdoctoral Program for Innovative Talents), the Beijing Natural Science Foundation under Grant Number 4244098, the Science Foundation of the Chinese Academy of Sciences, and the Swiss National Science Foundation (SNSF) project 200021E_219943 Neuromorphic Attention Models for Event Data (NAMED).

## Impact Statement

This paper presents work whose goal is to advance the field of efficient video-generation diffusion transformers. There are a few potential societal consequences of our work, none of which we feel must be specifically highlighted here.

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

# A. Proof of Theorem 3.2

**Lemma A.1.** *For any surjection $f : X \to Y$, we have*

$$\mathcal{H}(Y) \leq \mathcal{H}(X) \tag{20}$$

*Proof of Lemma A.1.*

For any surjection $f : X \to Y$, we have $p(y) = \sum_{f(x)=y} p(x)$. Thus, we have

$$
\begin{aligned}
\mathcal{H}(Y) &= -\sum_{y \in Y} p(y) \log p(y) \\
&= -\sum_{p(y)=p(x)} p(y) \log p(y) - \sum_{p(y) \neq p(x)} p(y) \log p(y) \\
&= -\sum_{p(y)=p(x)} p(y) \log p(y) - \sum_{p(y) \neq p(x)} \left( \sum_{x=f^{-1}(y)} p(x) \log \left( \sum_{x=f^{-1}(y)} p(x) \right) \right) \\
&= -\sum_{p(f(x))=p(x)} p(x) \log p(x) - \sum_{p(f(x)) \neq p(x)} p(x) \log \left( \sum_{z=f^{-1}(f(x))} p(z) \right) \\
&\leq -\sum_{p(f(x))=p(x)} p(x) \log p(x) - \sum_{p(f(x)) \neq p(x)} p(x) \log p(x) \\
&= \mathcal{H}(X)
\end{aligned}
\tag{21}
$$

Therefore, Lemma A.1 holds.

*Proof of Theorem 3.2.*

To simplify the proof, we omit $z$ in the quantization process of Eq. (3) and Eq. (4). Therefore, for any weight quantization error $\Delta$, we have

$$
\begin{aligned}
\mathcal{H}(\Delta) &= \mathcal{H}(\mathbf{W} - \hat{Q}(\mathbf{W})) \\
&= \mathcal{H}(\mathbf{W} - s\lfloor \frac{\mathbf{W}}{s} \rceil) \\
&= \mathcal{H}(s\frac{\mathbf{W}}{s} - s\lfloor \frac{\mathbf{W}}{s} \rceil) \\
&= \mathcal{H}(\frac{\mathbf{W}}{s} - \lfloor \frac{\mathbf{W}}{s} \rceil)
\end{aligned}
\tag{22}
$$

where $(\frac{\mathbf{W}}{s} - \lfloor \frac{\mathbf{W}}{s} \rceil)$ denotes only truncating the decimal part of $\frac{\mathbf{W}}{s}$. For any item $\mathbf{W}_i$ and $\mathbf{W}_j$ in $\mathbf{W}$, we have

$$
\begin{cases}
(\frac{\mathbf{W}_i}{s} - \lfloor \frac{\mathbf{W_i}}{s} \rceil) = (\frac{\mathbf{W}_j}{s} - \lfloor \frac{\mathbf{W_j}}{s} \rceil), & if \ \mathbf{W}_i = \mathbf{W}_j, \\
(\frac{\mathbf{W}_i}{s} - \lfloor \frac{\mathbf{W_i}}{s} \rceil) = (\frac{\mathbf{W}_j}{s} - \lfloor \frac{\mathbf{W_j}}{s} \rceil), & if \ \mathbf{W}_i \neq \mathbf{W}_j \ and \ (\frac{\mathbf{W}_i}{s} - \lfloor \frac{\mathbf{W_i}}{s} \rceil) = (\frac{\mathbf{W}_j}{s} - \lfloor \frac{\mathbf{W_j}}{s} \rceil), \\
(\frac{\mathbf{W}_i}{s} - \lfloor \frac{\mathbf{W_i}}{s} \rceil) \neq (\frac{\mathbf{W}_j}{s} - \lfloor \frac{\mathbf{W_j}}{s} \rceil), & if \ \mathbf{W}_i \neq \mathbf{W}_j \ and \ (\frac{\mathbf{W}_i}{s} - \lfloor \frac{\mathbf{W_i}}{s} \rceil) \neq (\frac{\mathbf{W}_j}{s} - \lfloor \frac{\mathbf{W_j}}{s} \rceil),
\end{cases}
\tag{23}
$$

Obviously, for mapping $g : \mathbf{W} \to \frac{\mathbf{W}}{s} - \lfloor \frac{\mathbf{W}}{s} \rceil$, $g$ is a surjection. Therefore, using Lemma A.1, we have

$$\mathcal{H}(\Delta) = \mathcal{H}(\frac{\mathbf{W}}{s} - \lfloor \frac{\mathbf{W}}{s} \rceil) \leq \mathcal{H}(\mathbf{W}) \tag{24}$$

Therefore, Theorem 3.2 holds.

*Table 6.* Performance of text-to-video generation on UCF-101 Dataset.

| Method | Bit-width (W/A) | FVD($\downarrow$) | FVD-FP16($\downarrow$) | CLIPSIM | CLIP-T | VQA-Aesthetic | VQA-Technical | $\Delta$ FLOW Score.($\downarrow$) | Temp. Flick. |
|---|---|---|---|---|---|---|---|---|---|
| - | 16/16 | 99.90 | 0.00 | 0.1970 | 0.9963 | 36.33 | 91.23 | 3.37 | 96.22 |
| Naive-PTQ | 4/6 | 197.85 | 244.06 | 0.1702 | 0.9921 | 4.98 | 0.36 | 66.51 | 63.72 |
| ViDiT-Q | 4/6 | 96.72 | 80.59 | 0.1940 | 0.9968 | 20.25 | 31.73 | 3.29 | 94.86 |
| **Q-VDiT** | 4/6 | 95.51 | 79.17 | 0.1944 | 0.9971 | 23.62 | 35.76 | 2.70 | 95.46 |
| Naive-PTQ | 3/8 | 220.83 | 251.74 | 0.1698 | 0.9913 | 3.23 | 0.17 | 69.06 | 61.42 |
| ViDiT-Q | 3/8 | 99.74 | 84.28 | 0.1925 | 0.9957 | 17.32 | 20.56 | 4.34 | 92.51 |
| **Q-VDiT** | 3/8 | 97.07 | 80.14 | 0.1935 | 0.9966 | 21.74 | 30.23 | 3.08 | 94.37 |
| Naive-PTQ | 3/6 | 221.09 | 252.37 | 0.1693 | 0.9910 | 3.16 | 0.16 | 69.85 | 61.14 |
| ViDiT-Q | 3/6 | 100.05 | 84.79 | 0.1923 | 0.9955 | 16.83 | 19.61 | 4.42 | 92.08 |
| **Q-VDiT** | 3/6 | 98.15 | 82.09 | 0.1934 | 0.9963 | 19.79 | 25.81 | 3.46 | 93.13 |

## B. Implementation Details

We follow the setup used in ViDiT-Q (Zhao et al., 2024) for different layer bits. We only quantize linear weight and use channel-wise quantization for weight, and dynamic token-wise quantization for activation like ViDiT-Q. We apply the same quantization setting for different methods for a fair comparison. We follow the quantization scheme used in PTQ4DiT (Wu et al., 2024a) and EfficientDM (He et al., 2023) to perform post-training calibration which fine-tunes the quantization parameters. For post-training quantization, we calibrate 5k iters for 6-8 bit, 10k iters for 4-bit, and 15k iters for 3-bit. For calibration parameters, we use a batch size of 4, learning rate of 1e-6 for weight quantization parameters, and 1e-5 for TQE parameters. We apply the same setting for other post-training-based methods. For calibration dataset, we use 10 prompts provided by Open-Sora (HPC-AI, 2024) and uniformly select 50 steps as used in ViDiT-Q (Zhao et al., 2024).

## C. Experimental Settings

For the Latte (Ma et al., 2024) Model, due to the lack of ground-truth videos for prompt-only datasets, we also report FVD-FP16 which chooses the FP16-generated video as ground-truth. All metrics are evaluated on 101 prompts (1 for each class) for UCF-101 (Soomro, 2012).

## D. Detailed Description of Selected Evaluation Metrics

### D.1. Benchmark Evaluation

Following VBench (Huang et al., 2024b) and previous work ViDiT-Q (Zhao et al., 2024), we select 8 dimensions from three key aspects in video-generation task.

1. **Frame-wise Quality.** In this aspect, we assess the quality of each individual frame without taking temporal quality into concern.
   - **Imaging Quality** assesses distortion (e.g., over-exposure, noise) presented in the generated frames using the MUSIQ (Ke et al., 2021) image quality predictor trained on the SPAQ (Fang et al., 2020) dataset.
   - **Aesthetic Quality** evaluates the artistic and beauty value perceived by humans towards each video frame using the LAION aesthetic predictor (LAION-AI, 2022).

2. **Temporal Quality.** In this aspect, we assess the cross-frame temporal consistency and dynamics.
   - **Dynamic Degree** evaluates the degree of dynamics (i.e., whether it contains large motions) generated by each model.
   - **Motion Smoothness** evaluates whether the motion in the generated video is smooth, and follows the physical law of the real world.
   - **Subject Consistency** assesses whether the subject's appearance remains consistent throughout the whole video.
   - **Background Consistency** evaluate the temporal consistency of the background scenes by calculating CLIP (Radford et al., 2021) feature similarity across frames.

3. **Semantics.** In this aspect, we evaluate the video's adherence to the text prompt given by the user. consistency.

- **Scene** evaluates whether the synthesized video is consistent with the intended scene described by the text prompt.
- **Overall Consistency** further use overall video-text consistency computed by ViCLIP (Wang et al., 2023) on general text prompts as an aiding metric to reflect both semantics and style consistency.

We use three different prompts set provided by the official github repository of VBench to generate videos. We generate one video for each prompt for evaluation same as ViDiT-Q (Zhao et al., 2024).

- **overall consistency.txt:** includes 93 prompts, used to evaluate overall consistency, aesthetic quality and imaging quality.

- **subject consistency.txt:** includes 72 prompts, used to evaluate subject consistency, dynamic degree, and motion smoothness.

- **scene.txt:** includes 86 prompts, used to evaluate scene and background consistency.

### D.2. Multi-aspects Metrics Evaluation

**CLIPSIM and CLIP-Temp:** CLIPSIM computes the image-text CLIP similarity for all frames in the generated videos and we report the averaged results. This quantifies the similarity between input text prompts and generated videos. CLIP-Temp computes the CLIP similarity of each two consecutive frames of the generated videos and then gets the averages on each two frames. This quantifies the semantics consistency of generated videos. We use the CLIP-VIT-B/32 (Wang et al., 2023) model to compute CLIPSIM and CLIP-Temp. We use the implementation from EvalCrafter (Liu et al., 2024) to compute these two metrics.

**DOVER's VQA:** VQA-Technical measures common distortions like noise, blur, and over-exposure. VQA-Aesthetic reflects aesthetic aspects such as the layout, the richness and harmony of colors, the photo-realism, naturalness, and artistic quality of the frames. We use the Dover (Wu et al., 2023) method to compute these two metrics.

**$\Delta$ FLOW Score:** Flow score wae proposed in (Liu et al., 2024) to measure the general motion information of the video. We use RAFT (Teed & Deng, 2020), to extract the dense flows of the video in every two frames and we calculate the average flow on these frames to obtain the average flow score of each generated video. In practice and in previous work (Zhao et al., 2024) finds, some poorly performing methods can cause the generated video to crash, resulting in an abnormally high FLOW Score. Thus, the difference Flow Score between the quantized Model and the FP Model is used as $\Delta$ FLOW Score for better comparison.

**Warping Error:** Warping error first obtain the optical flow of each two frames using the pre-trained optical flow estimation network (Teed & Deng, 2020). Then calculate the pixel-wise differences between the warped image and the predicted image. We calculate the warp differences on every two frames and calculate the final score using the average of all the pairs.

**FVD and FVD-FP16:** FVD measures the similarity between the distributions of features extracted from real and generated videos. We employ one randomly selected video per label from the UCF-101 dataset (101 videos in total) (Soomro, 2012) as the reference ground-truth videos for FVD evaluation. We follow (Zhao et al., 2024; Blattmann et al., 2023) to use a pretrained I3D model to extract features from the videos. Lower FVD scores indicate higher quality and more realistic video generation. However, due to relatively smaller video size (e.g. 101 videos in our case), employing FVD to evaluate video generation models faces several limitations. Small sample size cannot adequately represent either the diversity of the entire dataset or the complexity and nuances of video generation, leading to inaccurate and unstable results. To mitigate the limitations above, we propose an enhanced metric, FVD-FP16, for assessing the semantic loss in videos generated by quantized models relative to those produced by pre-quantized models. Specifically, we utilize 101 videos generated by the FP16 model as ground-truth reference videos. The FVD-FP16 has a significantly higher correlation with human perception.

**Temporal Flickering:** Temporal flickering measures temporal consistency at local and high-frequency details of generated videos. Then, we calculate the average MAE (mean absolute difference) value between each frame. We use the implementation in VBench (Huang et al., 2024b) to calculate temporal flickering.

## E. Additional Results on Latte Model

In Tab. 6, we present different quantization results about Latte (Ma et al., 2024) model on UCF-101 (Soomro, 2012) dataset. Compared to main baseline ViDiT-Q, our Q-VDiT still outperforms across all selected metrics under these harder bit settings. Under the hardest W3A6 setting, Q-VDiT still achieves VQA-Aesthetic of 19.79 and VQA-Technical of 25.81 which surpass the current method ViDiT-Q of 16.83 and 19.61.

*Table 7.* The illustration of Q-VDiT's hardware resource savings.

| Method | Bit-Width (W/A) | Memory Cost | Latency Cost | VQA-Aesthetic | VQA-Technical |
|---|---|---|---|---|---|
| - | 16/16 | 1.00× | 1.00× | 66.91 | 53.49 |
| ViDiT-Q | 4/8 | 2.42× | 1.38× | 60.62 | 49.38 |
| **Q-VDiT** | 4/8 | 2.40× | 1.35× | **71.32** | **55.56** |

## F. Latency Analysis of TQE

Introduced *Token-aware Quantization Estimator (TQE)* in Sec. 3.2 will bring extra computation cost in the actual quantization process. This technique is equivalent to a LoRA (Hu et al., 2021) module with $rank = 1$. SVDQuant (Li et al., 2024a) finds that the main bottleneck comes from memory access. Thus, SVDQuant proposes LORUNNER Kernel which fuses the down projection with the quantization kernel and the up projection with the quantization computation kernel, the low-rank branch can share the activations with the low-bit branch, eliminating the extra memory access and also halving the number of kernel calls. As a result, the low-rank branch adds only 5% latency with $rank = 16$, making it nearly cost-free. In our practice, TQE is only equivalent to $rank = 1$ which brings less latency burden. In Tab. 7, we present the memory cost and inference latency on W4A8 Open-Sora model. Compared to the full-precision model, our Q-VDiT can still bring 2.40× memory saving and 1.35× inference acceleration.

## G. More Qualitative Results

In the following pages, we visualize more generative video comparisons across different quantization methods. For better comparison, we uniformly sample 8 frames of each video.

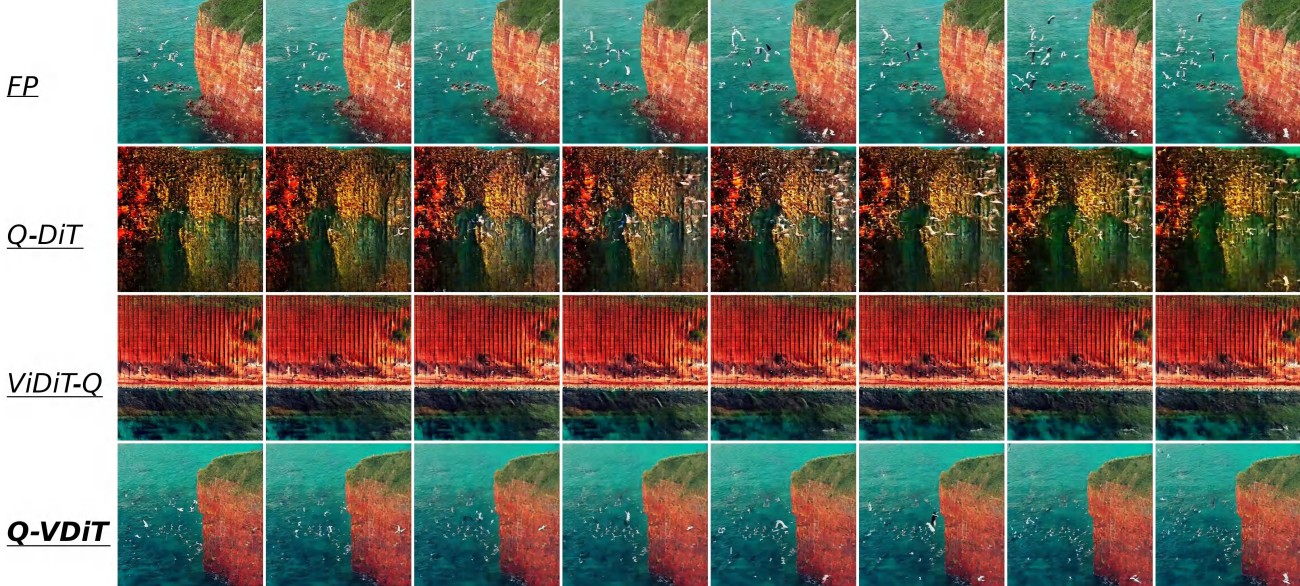

*Figure 7.* The qualitative results with prompt "A soaring drone footage captures the majestic beauty of a coastal cliff, its red and yellow stratified rock faces rich in color and against the vibrant turquoise of the sea. Seabirds can be seen taking flight around the cliff's precipices. As the drone slowly moves from different angles, the changing sunlight casts shifting shadows that highlight the rugged textures of the cliff and the surrounding calm sea. The water gently laps at the rock base and the greenery that clings to the top of the cliff, and the scene gives a sense of peaceful isolation at the fringes of the ocean. The video captures the essence of pristine natural beauty untouched by human structures.".

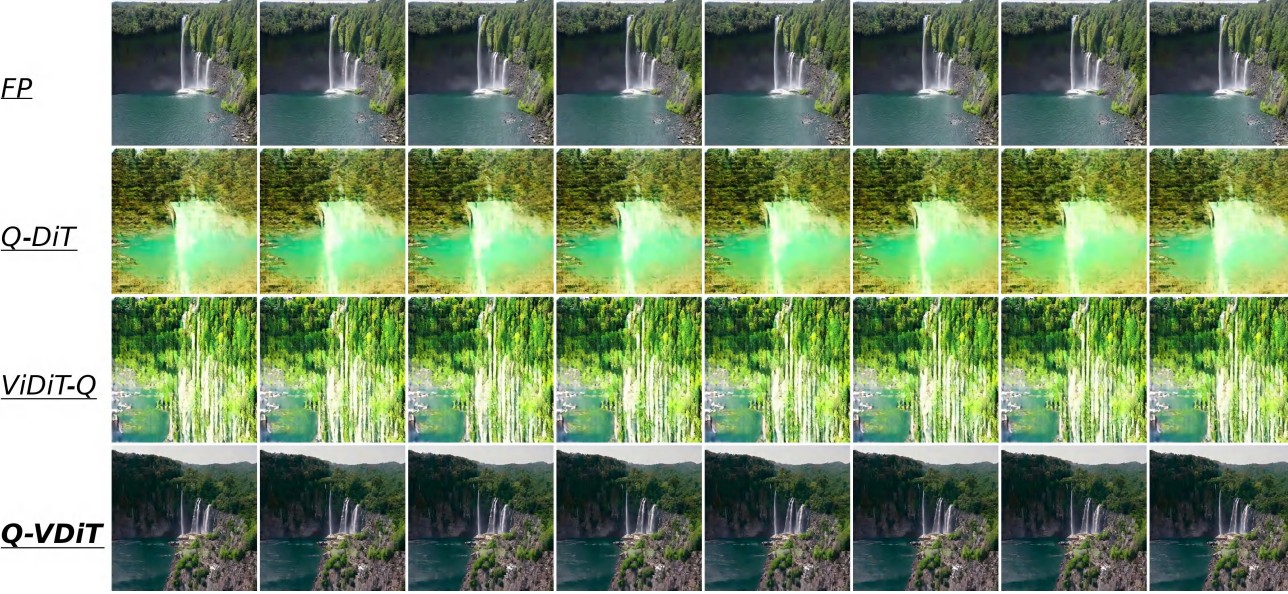

*Figure 8.* The qualitative results with prompt "The video captures the majestic beauty of a waterfall cascading down a cliff into a serene lake. The waterfall, with its powerful flow, is the central focus of the video. The surrounding landscape is lush and green, with trees and foliage adding to the natural beauty of the scene. The camera angle provides a bird's eye view of the waterfall, allowing viewers to appreciate the full height and grandeur of the waterfall. The video is a stunning representation of nature's power and beauty.".

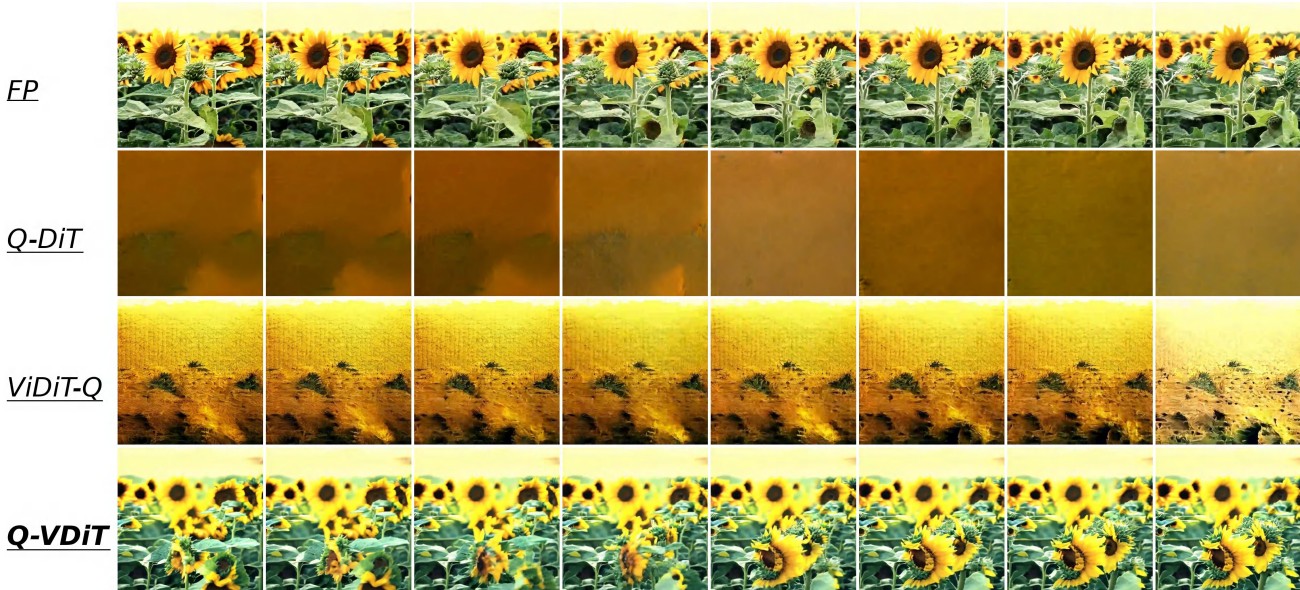

*Figure 9.* The qualitative results with prompt "The vibrant beauty of a sunflower field. The sunflowers, with their bright yellow petals and dark brown centers, are in full bloom, creating a stunning contrast against the green leaves and stems. The sunflowers are arranged in neat rows, creating a sense of order and symmetry. The sun is shining brightly, casting a warm glow on the flowers and highlighting their intricate details. The video is shot from a low angle, looking up at the sunflowers, which adds a sense of grandeur and awe to the scene. The sunflowers are the main focus of the video, with no other objects or people present. The video is a celebration of nature's beauty and the simple joy of a sunny day in the countryside.".

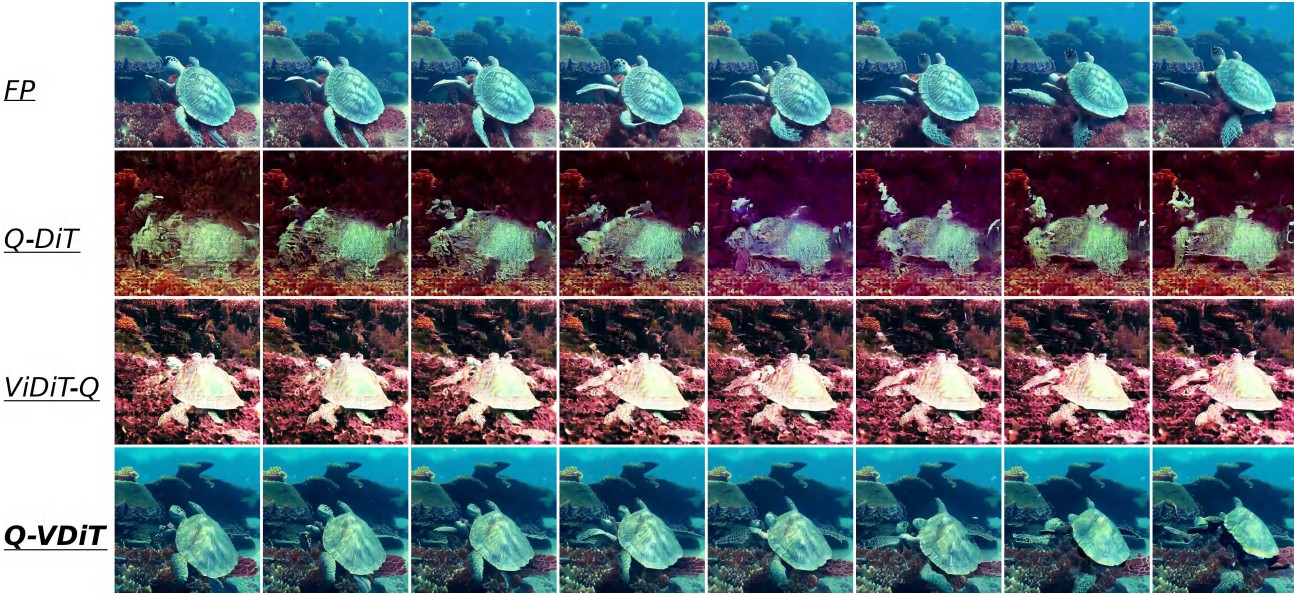

*Figure 10.* The qualitative results with prompt "A serene underwater scene featuring a sea turtle swimming through a coral reef. The turtle, with its greenish-brown shell, is the main focus of the video, swimming gracefully towards the right side of the frame. The coral reef, teeming with life, is visible in the background, providing a vibrant and colorful backdrop to the turtle's journey. Several small fish, darting around the turtle, add a sense of movement and dynamism to the scene. The video is shot from a slightly elevated angle, providing a comprehensive view of the turtle's surroundings. The overall style of the video is calm and peaceful, capturing the beauty and tranquility of the underwater world.".

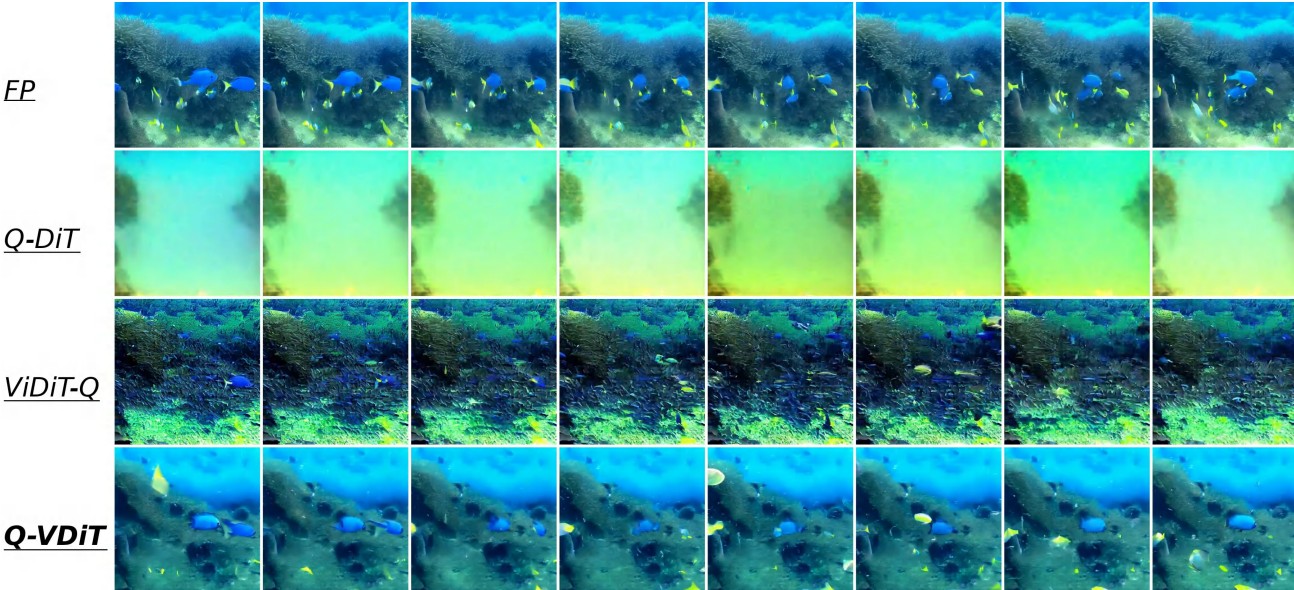

*Figure 11.* The qualitative results with prompt "A vibrant underwater scene. A group of blue fish, with yellow fins, are swimming around a coral reef. The coral reef is a mix of brown and green, providing a natural habitat for the fish. The water is a deep blue, indicating a depth of around 30 feet. The fish are swimming in a circular pattern around the coral reef, indicating a sense of motion and activity. The overall scene is a beautiful representation of marine life.".

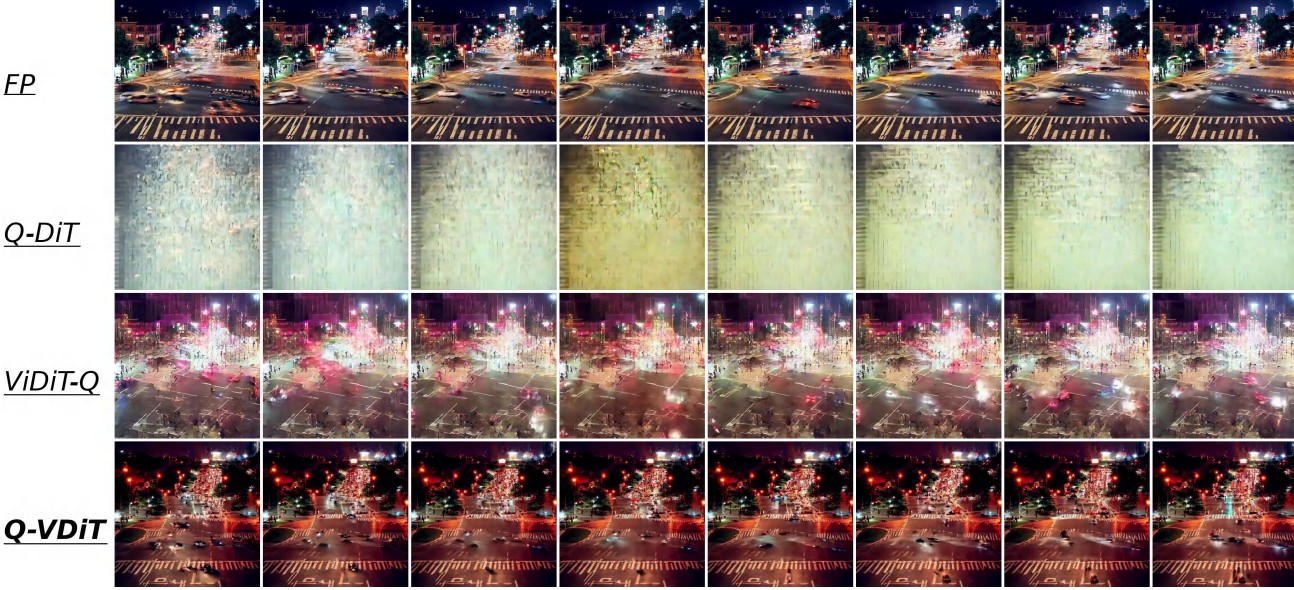

*Figure 12.* The qualitative results with prompt "A bustling city street at night, filled with the glow of car headlights and the ambient light of streetlights. The scene is a blur of motion, with cars speeding by and pedestrians navigating the crosswalks. The cityscape is a mix of towering buildings and illuminated signs, creating a vibrant and dynamic atmosphere. The perspective of the video is from a high angle, providing a bird's eye view of the street and its surroundings. The overall style of the video is dynamic and energetic, capturing the essence of urban life at night.".

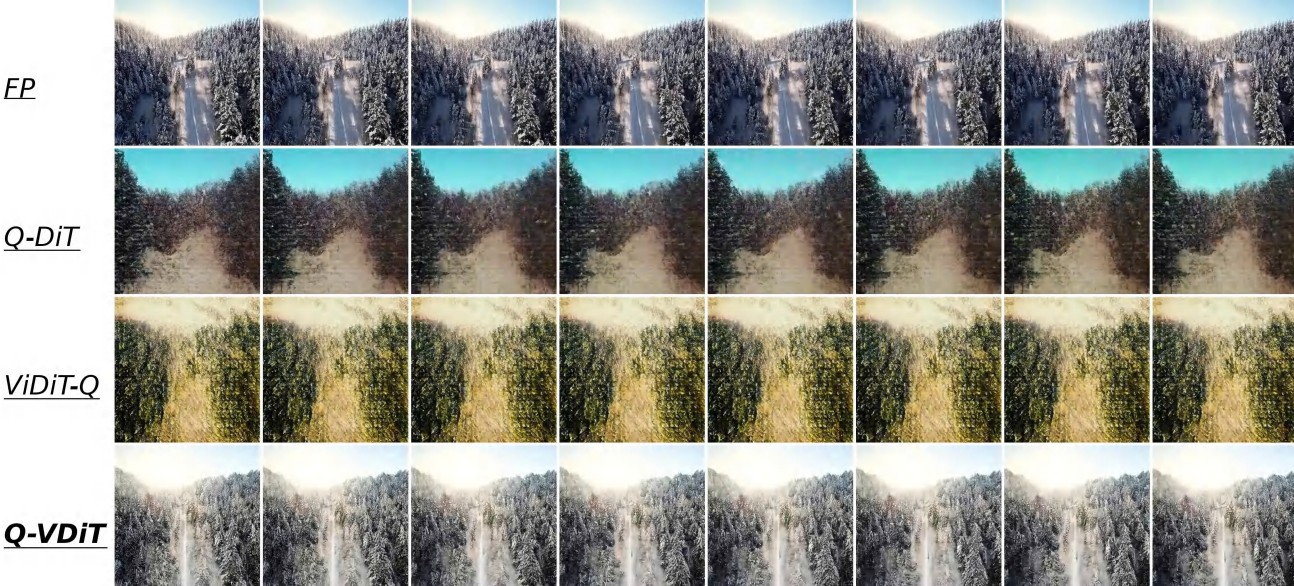

*Figure 13.* The qualitative results with prompt "A snowy forest landscape with a dirt road running through it. The road is flanked by trees covered in snow, and the ground is also covered in snow. The sun is shining, creating a bright and serene atmosphere. The road appears to be empty, and there are no people or animals visible in the video. The style of the video is a natural landscape shot, with a focus on the beauty of the snowy forest and the peacefulness of the road.".

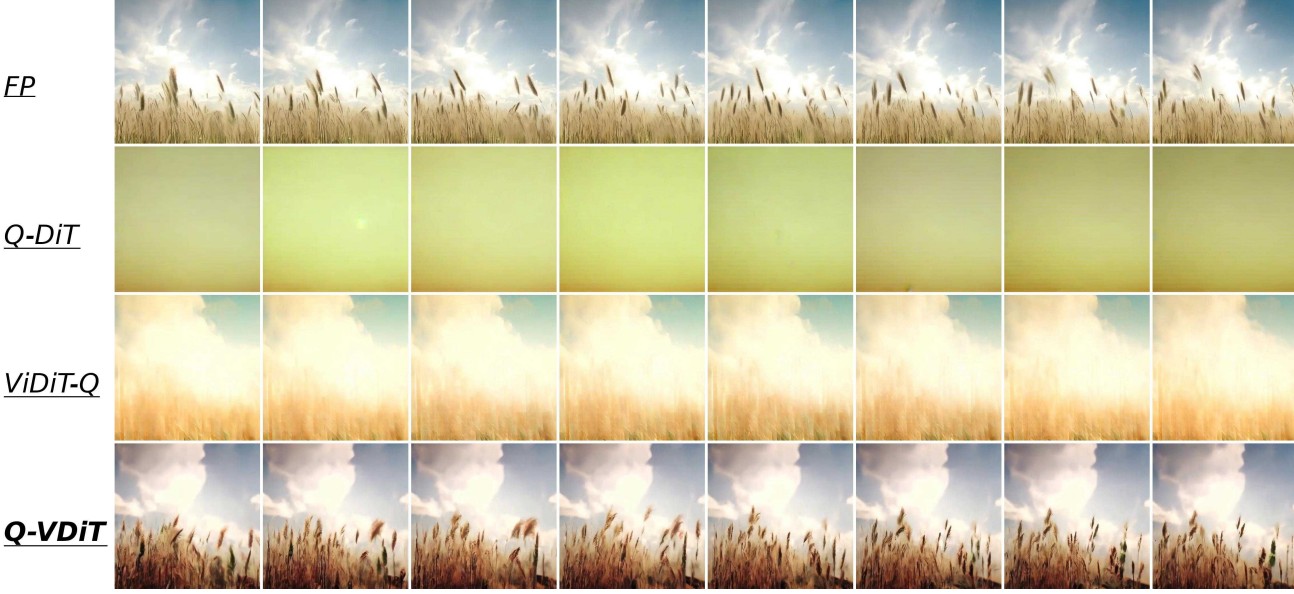

*Figure 14.* The qualitative results with prompt "The dynamic movement of tall, wispy grasses swaying in the wind. The sky above is filled with clouds, creating a dramatic backdrop. The sunlight pierces through the clouds, casting a warm glow on the scene. The grasses are a mix of green and brown, indicating a change in seasons. The overall style of the video is naturalistic, capturing the beauty of the landscape in a realistic manner. The focus is on the grasses and their movement, with the sky serving as a secondary element. The video does not contain any human or animal elements.".

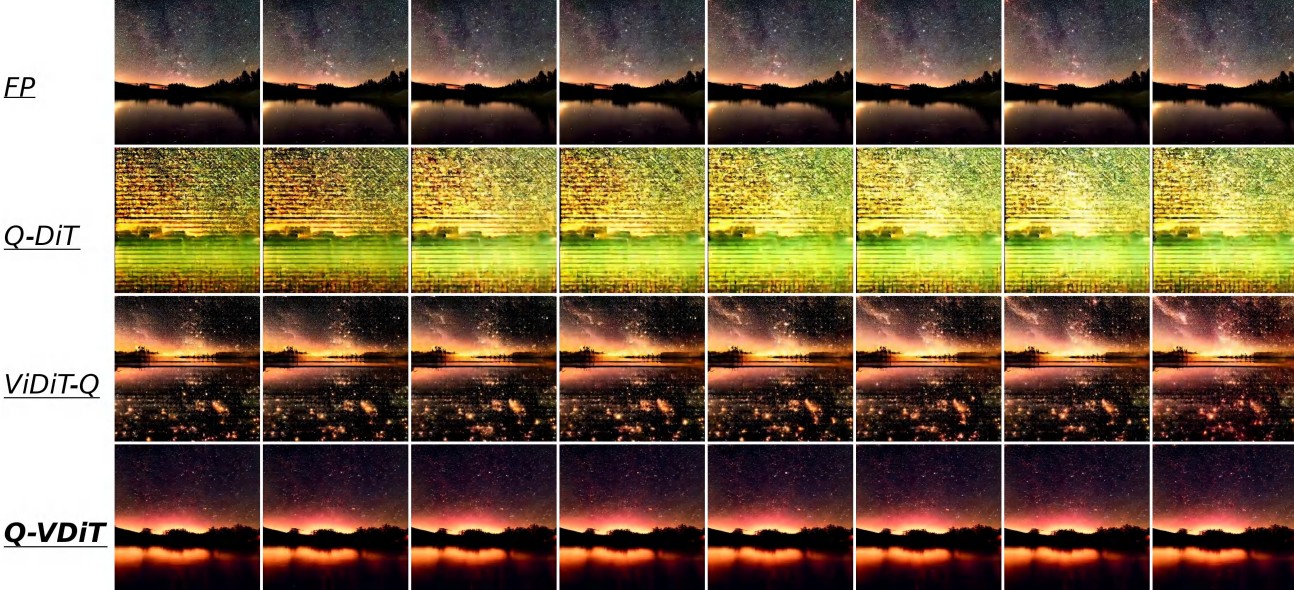

*Figure 15.* The qualitative results with prompt "A serene night scene in a forested area. The first frame shows a tranquil lake reflecting the star-filled sky above. The second frame reveals a beautiful sunset, casting a warm glow over the landscape. The third frame showcases the night sky, filled with stars and a vibrant Milky Way galaxy. The video is a time-lapse, capturing the transition from day to night, with the lake and forest serving as a constant backdrop. The style of the video is naturalistic, emphasizing the beauty of the night sky and the peacefulness of the forest.".

