# OpenReview forum: "Q-VDiT: Towards Accurate Quantization and Distillation of Video-Generation Diffusion Transformers"
_ICML.cc/2025/Conference — ICML 2025 poster_

### Official Review · Reviewer_Xj8f · 2025-03-11

**Overall Recommendation:** 2

**Summary:**

This paper proposed a quantization method named Q-VDiT tailored specifically for video Diffusion Transformers. The proposed Q-VDiT aims to address severe model quantization information loss in video models. Specifically, the authors proposed Token-aware Quantization Estimator to compensate for quantization errors from both token and feature dimensions. Temporal Maintenance Distillation is used to optimize each frame from the perspective of the overall video. Extensive experiments demonstrate the superiority of the proposed Q-VDiT over baseline and other previous quantization methods.

**Claims And Evidence:**

This paper claims that "Quantization can reduce storage requirements and accelerate inference by lowering the bit-width of model parameters. Yet, existing quantization methods for image generation models do not generalize well to video generation tasks.". The claims are supported by experimental results.

**Essential References Not Discussed:**

No

**Experimental Designs Or Analyses:**

Yes. Quantization Estimator to compensate for quantization errors from both token and feature dimensions. Temporal Maintenance Distillation is used to optimize each frame from the perspective of the overall video.

**Methods And Evaluation Criteria:**

The authors proposed Token-aware Quantization Estimator to compensate for quantization errors from both token and feature dimensions. Temporal Maintenance Distillation is used to optimize each frame from the perspective of the overall video. Extensive experiments demonstrate the superiority of the proposed Q-VDiT over baseline and other previous quantization methods.

**Other Comments Or Suggestions:**

My major concern is the unsatisfied video generation performance.

**Other Strengths And Weaknesses:**

Pros:

1.	a quantization method named Q-VDiT tailored specifically for video Diffusion Transformers is proposed.
2.	Token-aware Quantization Estimator is proposed to compensate for quantization errors from both token and feature dimensions.
3.	Temporal Maintenance Distillation is used to optimize each frame from the perspective of the overall video
4.	Experimental results verify the effectiveness of the proposed method.

Cons:

1.	More optimization details of Temporal Maintenance Distillation should be included such as training data and resources.
2.	The video demos generated by the proposed method, included in the supplementary material, show observed artifacts and temporal flicking, the authors should give some deeper explanations.
3.	The original videos generated by the open-sora model are not displayed, and it seems that the proposed  Quantization method can not preserve the performance of the original models.

**Questions For Authors:**

Please refer to Weaknesses for more details.

**Relation To Broader Scientific Literature:**

This paper proposed a quantization method named Q-VDiT tailored specifically for video Diffusion Transformers. The proposed Q-VDiT aims to address severe model quantization information loss in video models. Specifically, the authors proposed Token-aware Quantization Estimator to compensate for quantization errors from both token and feature dimensions. Temporal Maintenance Distillation is used to optimize each frame from the perspective of the overall video. Extensive experiments demonstrate the superiority of the proposed Q-VDiT over baseline and other previous quantization methods.

**Theoretical Claims:**

Yes. Proof of Theorem 3.2 is reasonable.

---

> ### Author Rebuttal · Authors · 2025-04-01
>
> We thank the reviewer for the constructive comments on our paper. Regarding the concerns, we provide the following responses.
> > Q1: Optimization details.
>
> Sorry for the misunderstanding, **we have reported optimization details including training data in Appendix Sec. B. and show the training cost in Tab. 4**. We will make this more prominent in revised version.
> > Q2: Original videos.
>
> We apologize for our negligence and we have released all original videos in [[https://anonymous.4open.science/r/Generated_videos]](https://anonymous.4open.science/r/Generated_videos-EB77). **We also have shown more generated critical video frames under W3A6 setting comparison with FP model in Appendix Sec. H. In W3A6 setting, current methods cannot even produce meaningful videos. Our method is significantly closer to FP in terms of generation quality than existing methods.**
> > Q3: Some temporal flicking.
>
> The OpenSora model we experimented in paper inevitably has some temporary flicking. But our method is significantly better than existing methods in terms of metrics (see Tab. 1 and Tab. 2) and closer to the FP model in terms of visual effects (see Fig. 5, Appendix Sec. H). **We further conducted W4A6 quantization experiments on larger models HunyuanVideo and CogVideoX, and demonstrated the generated video effects in [[https://anonymous.4open.science/r/Generated_videos]](https://anonymous.4open.science/r/Generated_videos-EB77). The videos we generate do not have issues such as temporary flicking and the visual quality is closer to the full precision model. We also has notable improvement compared to the baseline method ViDiT-Q.**
>
> We also present quantitative comparisons in the table below：
>
> |Model|Method|Imaging Quality($\uparrow$)|Aesthetic Quality($\uparrow$)|Overall Consistency($\uparrow$)|
> |-|-|-|-|-|
> |CogVideoX|FP|61.80|58.88|26.46|
> |CogVideoX|ViDiT-Q|46.03|45.31|21.65|
> |CogVideoX|**Ours**|**52.13**|**49.86**|**23.75**|
> |Hunyuan|FP|62.30|62.49|26.85|
> |Hunyuan|ViDiT-Q|52.28|55.25|24.81|
> |Hunyuan|**Ours**|**57.42**|**57.04**|**25.49**|
>
> Our method is closer to the FP model in terms of metrics and visual effects, and has a significant improvement compared to the baseline method ViDiT-Q.
>
> > Q4: Performance of our quantized model compared with FP model.
>
> **1. Some performance gap under lower bit setting.** We have reported the same higher bit setting (e.g. W6A6) with ViDiT-Q in Appendix Tab. 5 and our method can achieve lossless performance compared with FP model. Since the performance gap is minor in higher bit settings, we further explore the performance improvement on lower bit settings (e.g., W3A6). Under this low bit setting, we have achieved state-of-the-art and **significantly outperform existing methods and existing methods can hardly generate reasonable videos under low bit setting.** We show more visual comparison in Fig. 5 and Appendix Sec. H.
>
> **2. Choice for mainly focus on lower bit.** Naturally, lower bit quantization brings more memory saving and acceleration for real-world deployment, but often faces more severe performance degradation which is a harder situation under exploration. Since existing methods like ViDiT-Q have achieved almost lossless performance at higher bits (e.g., W4A8), we want to further explore the performance improvement at lower bits. Compared to W4 quantization, W3 usually faces severe performance degradation which is commonly discovered in LLM quantization [1][2]. So we chose lower bit quantization settings (e.g., W3A6) in Tab. 1 and Tab. 2, under which existing methods can hardly generate reasonable videos as shown in Fig. 5. Our method has greatly improved in terms of metrics and visual effects compared to existing methods. **We want to note that we also reported same higher bit settings as ViDiT-Q in Appendix Tab. 5**, and our method still has improvement compared to existing methods and **achieves lossless**.
>
> [1].GPTQ: Accurate Post-Training Quantization for Generative Pre-trained Transformers.
>
> [2].Quarot: Outlier-free 4-bit inference in rotated llms.
>
> **3. Main contribution.** We would like to further highlight our contribution by investigating the limitations of current quantization methods in video generation models and solving them from both quantization (Sec. 3.2) and optimization (Sec. 3.3) perspectives. With almost the same calibration cost with current methods (Tab. 4), our method brings significant relatively performance improvement at lower bit compared to existing methods in terms of metrics (Tab. 1 and Tab. 2) and visual effects (Fig. 5 and Appendix Sec. H) while maintaining lossless performance at higher bit (Appendix Tab. 5). Meanwhile, our method can bring 2.4x reduction in memory cost and 1.35x actual acceleration for inference (Appendix Tab. 7) with no extra burden compared with baseline ViDiT-Q.

---

### Official Review · Reviewer_ZxGk · 2025-03-13

**Overall Recommendation:** 3

**Summary:**

This paper addresses the issue of information loss and misalignment of optimization objectives that arise when applying existing quantization methods to video generation models. Current quantization techniques, which are primarily designed for image generation models, may not be directly suitable for video generation due to the temporal dependencies inherent in videos. To address this challenge, the paper proposes a Token-aware Quantization Estimator (TQE) and a Token-aware Quantization Estimator (TQE). The former aims to reduce information loss by considering token-wise importance during quantization, while the latter focuses on minimizing the temporal distribution discrepancy between the full-precision model and the quantized model. Experimental results demonstrate that the proposed approach effectively mitigates information loss compared to existing quantization methods when applied to video generation models.

**Claims And Evidence:**

yes

**Essential References Not Discussed:**

No

**Experimental Designs Or Analyses:**

yes, I have reviewed all the experiments presented in the paper, which are all necessary. However, the analysis of the experiments is not sufficiently detailed.

1.	Although the proposed method outperforms state-of-the-art methods in terms of final generation metrics, it remains unclear whether the improvements genuinely stem from the reduction in quantization error or from other factors. Further analysis is needed to verify the source of the performance gains.
2.	In the experiment, the paper mainly compares the video generation performance at different bit-widths but does not provide a thorough analysis of the acceleration effect or memory savings. This lack of evaluation makes it difficult to assess the actual efficiency improvements brought by the proposed method.
3.	In Table.4, as ViDiT-Q is a calibration-free method, why does it still have training cost? In this table, it would be better to show the memory consumption of the full-precision and the fp16 models, which would help demonstrate the efficiency of the proposed method.

**Methods And Evaluation Criteria:**

yes

**Other Comments Or Suggestions:**

In Figure 2, some symbols or notations appear to be missing, which may affect the clarity and completeness of the illustration. It would be helpful to ensure all necessary symbols are properly displayed.

**Other Strengths And Weaknesses:**

Strengths:
1.	The writing in the paper is relatively clear, making it easy to follow the motivation and the proposed method.
2.	The proposed method outperforms the state-of-the-art methods on public datasets. Additionally, the ablation study validates the effectiveness of each component.

 Weaknesses:
please see the comments and questions in different sections.

**Questions For Authors:**

What similarity function is used in Eq. 9

**Relation To Broader Scientific Literature:**

Current quantization methods are primarily designed for image generation models, whereas the method proposed in this paper introduces a quantization approach specifically for compressing and accelerating video generation models.

**Theoretical Claims:**

yes, the formulas and proofs seem correct.

---

> ### Author Rebuttal · Authors · 2025-04-01
>
> Thank you very much for your high recognition of our work and the valuable suggestions you provided. Our response is as follows:
> > Q1: Quantitative analysis on quantization error.
>
> We add quantitative experiments on W3A6 model last layer weight's  quantization error and information entropy mentioned in the proposed TQE (Sec. 3.2)：
>
> |Method|Quantization Error($\downarrow$)|Entropy($\uparrow$)|VQA-Aesthetic($\uparrow$)|
> |-|-|-|-|
> |FP|-|6.98|66.91|
> |ViDiT-Q|73.7|4.46|39.82|
> |**Ours**|**56.0**|**5.49**|**53.53**|
>
> **Consistent with our claim, our method indeed reduces quantization errors and improves the information entropy of the quantized weights.** From quantitative metrics, TQE has indeed improved the performance of the model by reducing quantization error and increasing entropy. Theorem 3.2 states that the quantization error of weights has lower information entropy compared to the original weights. This is also proved by the quantitative results.
>
> > Q2: Actual inference efficiency and memory consumption.
>
> We have reported the inference efficiency of W4A8 model compared with FP model in Appendix Tab. 7. We also display the data here.
> |Method|Memory Cost($\downarrow$)|Latency Cost($\downarrow$)|VQA-Aesthetic($\uparrow$)|VQA-Technical($\uparrow$)|
> |-|-|-|-|-|
> |FP|10.9G|51.3s|66.91|53.49|
> |**Ours**|**4.5G (2.4$\times$)**|**38.0s (1.35$\times$)**|**71.32**|**55.56**|
>
> Compared to FP model, our method achieves complete lossless performance while bringing 2.4$\times$ reduction in memory cost and 1.35$\times$ inference acceleration. In the revised version, we will modify our layout to make this more prominent.
>
> > Q3: In Tab. 4, why ViDiT-Q has training cost?
>
>
> ViDiT-Q requires calculating quantization sensitivity between layers to allocate different bit-widths, which incurs additional time consumption. This is also reported in their paper. We will modify our wording to avoid ambiguity.
>
> > Q4: Fig.2 misses some notations.
>
> We will fix the problem in the final version.
>
> > Q5: Similarity function used in Eq. 9.
>
> We use cosine similarity.

---

### Official Review · Reviewer_WtXR · 2025-03-14

**Overall Recommendation:** 3

**Summary:**

The paper introduces Q-VDiT, a quantization framework for video DiT to reduce computational complexity while preserving video quality. It addresses two key challenges: quantization error compensation through a Token-aware Quantization Estimator (TQE) and spatiotemporal consistency via Temporal Maintenance Distillation (TMD).

## update after rebuttal
Thanks to the authors for addressing my concerns and providing additional results. I will keep my score. The videos on Hunyuan/opensora look great.

**Claims And Evidence:**

The statement that existing approaches fail to calibrate quantization from a video-wide perspective, leading to degraded video quality is too strong. ViDiT-Q does incorporate video tokens and reports results on video datasets, making it inaccurate to claim that it does not consider the entire video. The distinction should focus on how Q-VDiT improves over ViDiT-Q rather than dismissing prior work outright.

**Essential References Not Discussed:**

n/a

**Experimental Designs Or Analyses:**

- The paper adopts a different quantization setting from ViDiT-Q but lacks a clear justification. The claim that "we mainly focus on harder settings" does not sufficiently explain the reasoning behind the choice. Further insights into why these settings are chosen, particularly regarding their relevance to real-world deployment and model robustness, would improve the clarity of the paper.
- The paper does not explicitly discuss the performance gap between W3A8 and W4A8 (in ViDiT-Q). Given the significant differences in VBench scores between this paper and ViDiT-Q, a more detailed comparison is necessary. Clarifying whether W3A8 introduces substantial quality degradation compared to W4A8 would help readers interpret the reported results.

**Methods And Evaluation Criteria:**

The Token-aware Quantization Estimator (TQE) is introduced to approximate quantization errors across two orthogonal dimensions: token and feature space. However, its quantization error reduction is not directly evaluated. While Figure 3 provides an example and Table 3 includes an ablation study, the effectiveness of TQE in minimizing quantization error remains unclear. A more explicit quantitative evaluation of the error reduction would strengthen the argument.

**Other Comments Or Suggestions:**

n/a

**Other Strengths And Weaknesses:**

n/a

**Questions For Authors:**

- why the generated videos exhibit low motion dynamics? Do we observe this before using quantization?
- The paper does not evaluate on larger video generation models, such as Hunyuan (https://github.com/Tencent/HunyuanVideo), why? The results could be more meaningful on these SOTA models.

**Relation To Broader Scientific Literature:**

not related

**Theoretical Claims:**

Proposition 3.1 and Theorem 3.2 establish that quantized weights retain less information entropy than the original weights. However, the link between this theoretical result and the practical benefits of TQE is not clearly articulated. It remains unclear which specific error terms TQE reduces and by how much. Providing quantitative results on TQE’s impact on entropy reduction would help substantiate its contribution.

---

> ### Author Rebuttal · Authors · 2025-04-01
>
> Thank you for your detailed review of our work. Here are our responses to your concerns:
> > Q1: Statement of our Q-VDiT.
>
> We apologize for the misunderstanding caused by our statement. **We absolutely do not deny ViDiT-Q's contribution. ViDiT-Q is the first method to explore quantization for video generation models and an important baseline method for our paper. We greatly appreciate the contribution of ViDiT-Q.** What the statement wants to emphasize is that we hope to consider the correlation between different frames in the video information during the optimization process and further improve the quality of the quantized video generation model. We will modify our wording in the revised version to avoid this ambiguity.
>
> > Q2: Quantitative analysis on quantization error and information entropy introduced in TQE.
>
> We add quantitative experiments on W3A6 model last layer weight's  quantization error and information entropy mentioned in the proposed TQE (Sec. 3.2)：
>
> |Method|Quantization Error($\downarrow$)|Weight Entropy($\uparrow$)|VQA-Aesthetic($\uparrow$)|
> |-|-|-|-|
> |FP|-|6.98|66.91|
> |ViDiT-Q|73.7|4.46|39.82|
> |**Ours**|**56.0**|**5.49**|**53.53**|
>
> **Consistent with our claim, our method indeed reduces quantization errors and improves the information entropy of the quantized weights.** From quantitative metrics, TQE has indeed improved the performance of the model by reducing quantization error and increasing entropy. Theorem 3.2 states that the quantization error of weights has lower information entropy compared to the original weights. This is also proved by the quantitative results.
>
> > Q3: Explanation of quantization settings.
>
> Naturally, lower bit quantization brings more memory saving and acceleration for real-world deployment, but often faces more severe performance degradation, which is a harder situation under exploration. Since existing methods like ViDiT-Q have achieved almost lossless performance at higher bits (e.g., W4A8), we want to further explore the performance improvement at lower bits. Compared to W4 quantization, W3 usually faces severe performance degradation, which is commonly discovered in LLM quantization [1][2]. So we chose lower bit quantization settings (e.g., W3A6) in Tab. 1 and Tab. 2, under which existing methods can hardly generate reasonable videos as shown in Fig. 5. Our method has greatly improved in terms of metrics and visual effects compared to existing methods. **We want to note that we also reported the same higher bit settings as ViDiT-Q in Appendix Tab. 5**, and our method still has improvement compared to existing methods and **achieves lossless**.
>
> [1].GPTQ: Accurate Post-Training Quantization for Generative Pre-trained Transformers.
>
> [2].Quarot: Outlier-free 4-bit inference in rotated llms.
>
> > Q4: Performance gap between W3A8 and W4A8.
>
> Compared to W4, the significant loss of weight information in W3 (only 2\**3=8 representation candidates for a single value) leads to a serious performance degradation as we discussed in Q3. This also highlights the importance of our proposed TQE in compensating for the weight quantization errors. **Our method outperforms current quantization methods in both W3 and W4 (Tab. 1 and Tab. 2)**.
>
> > Q5: Motion dynamics before and after quantization.
>
> Existing methods like ViDiT-Q have also found that the quantized model exhibits a certain degree of dynamic decline. We can also see from the metrics Motion Smoothness and Dynamic Degree in Tab. 1 that this is particularly severe after W3 quantization. **Our method has the same level of motion dynamics as the full precision model in terms of metrics**, which can be seen from the Motion Smoothness and Dynamic Degree metrics in Tab. 1. And our method also shows significant improvement in dynamics compared to existing methods. More generated videos in [[https://anonymous.4open.science/r/Generated_videos]](https://anonymous.4open.science/r/Generated_videos-EB77) also prove the dynamic retention ability of our method at the same level as the FP model.
>
> > Q6: Evaluation on larger video generation models.
>
> We add W4A6 quantization experiment on larger SOTA models CogVideoX-5B and HunyuanVideo-13B:
>
> |Model|Method|Imaging Quality($\uparrow$)|Aesthetic Quality($\uparrow$)|Overall Consistency($\uparrow$)|
> |-|-|-|-|-|
> |CogVideoX|FP|61.80|58.88|26.46|
> |CogVideoX|ViDiT-Q|46.03|45.31|21.65|
> |CogVideoX|**Ours**|**52.13**|**49.86**|**23.75**|
> |Hunyuan|FP|62.30|62.49|26.85|
> |Hunyuan|ViDiT-Q|52.28|55.25|24.81|
> |Hunyuan|**Ours**|**57.42**|**57.04**|**25.49**|
>
> **We also provide more generated video comparisons in [[https://anonymous.4open.science/r/Generated_videos]](https://anonymous.4open.science/r/Generated_videos-EB77). Our method is closer to the FP model in terms of metrics and visual effects, and has notable improvement compared to the baseline method ViDiT-Q.**

---

### Official Review · Reviewer_paDY · 2025-03-16

**Overall Recommendation:** 3

**Summary:**

Diffusion transformers (DiT) are powerful for video generation but face deployment challenges due to large parameter sizes and high computational complexity. To tackle the issues of information loss and mismatched objectives during quantization, the authors propose Q-VDiT, introducing the Token aware Quantization Estimator (TQE) to mitigate quantization errors and the Temporal Maintenance Distillation (TMD) to preserve spatiotemporal correlations across frames. Their W3A6 Q-VDiT achieves a scene consistency score of 23.40, surpassing current state-of-the-art quantization methods by 1.9.

**Claims And Evidence:**

Yes. The claims are supported by clear and convincing evidence.

**Essential References Not Discussed:**

NA

**Experimental Designs Or Analyses:**

Yes, I verified that the experimental designs and analyses were sound. The main findings rely on high-level metrics.

**Methods And Evaluation Criteria:**

Make lots of sense but missing some love level comparisons.

**Other Comments Or Suggestions:**

NA

**Other Strengths And Weaknesses:**

Strengths:

The idea is both good and novel.

The quantitative results are notably strong.

The writing is easy to follow.

Weaknesses:

The supplementary material does not show the original model’s qualitative results, and overall quality does not seem improved compared to other methods.

Although the high-level metrics are strong, the presented video exhibits noticeable blurriness; reporting additional lower-level metrics (e.g., FVD) could provide a more comprehensive evaluation.

Inference speed comparisons are missing, making it difficult to assess practical efficiency.

The motivation for concatenating each frame with all frames for temporal distillation is unclear, especially given potential alternatives (e.g., temporal differences).

**Questions For Authors:**

Please see above parts

**Relation To Broader Scientific Literature:**

The paper makes a valuable contribution to the field of video generation by addressing the issue of large, slow models, and demonstrating how quantization can significantly speed up inference.

**Theoretical Claims:**

Yes, I have checked the proofs for theoretical claim regarding all equations.

---

> ### Author Rebuttal · Authors · 2025-04-01
>
> Thank you for reviewing our manuscript and providing valuable suggestions. Here are our responses to some of the concerns you raised:
> > Q1: Original model’s qualitative results.
>
> We apologize for our negligence and we have released all original videos in [[https://anonymous.4open.science/r/Generated_videos]](https://anonymous.4open.science/r/Generated_videos-EB77). We have also shown more generated critical video frames under W3A6 setting comparison with FP model in Appendix Sec. H. In W3A6 setting, current methods cannot even produce meaningful videos. Our method is significantly closer to FP model in terms of generation quality than existing methods.
>
> > Q2: Overall quality improvement.
>
> We have reported the same higher bit setting (e.g. W6A6) with ViDiT-Q in Appendix Tab. 5 and our method can achieve lossless performance compared with FP model. Since the performance gap is minor in higher bit settings, we further explore the performance improvement on lower bit settings (e.g., W3A6). Under this low bit setting, we have achieved state-of-the-art and **significantly outperform existing methods and existing methods can hardly generate reasonable videos under low bit setting.** We show more visual comparison in Fig. 5 and Appendix Sec. H.
>
> **We further conducted W4A6 quantization experiments on larger models HunyuanVideo and CogVideoX, and demonstrated the generated video effects in [[https://anonymous.4open.science/r/Generated_videos]](https://anonymous.4open.science/r/Generated_videos-EB77)**. The visual quality of our method is closer to the full precision model. We obtained better results than baseline method ViDiT-Q.
>
> We also present quantitative comparisons in the table below：
>
> |Model|Method|Imaging Quality($\uparrow$)|Aesthetic Quality($\uparrow$)|Overall Consistency($\uparrow$)|
> |-|-|-|-|-|
> |CogVideoX|FP|61.80|58.88|26.46|
> |CogVideoX|ViDiT-Q|46.03|45.31|21.65|
> |CogVideoX|**Ours**|**52.13**|**49.86**|**23.75**|
> |Hunyuan|FP|62.30|62.49|26.85|
> |Hunyuan|ViDiT-Q|52.28|55.25|24.81|
> |Hunyuan|**Ours**|**57.42**|**57.04**|**25.49**|
>
> Our method is closer to the FP model in terms of metrics and visual effects, and has a significant improvement compared to the baseline method ViDiT-Q.
>
> > Q3: Additional lower-level metrics (e.g., FVD).
>
> We have reported FVD on UCF-101 dataset in Appendix Tab. 6. We also report FVD on OpenSORA prompt set used in Tab. 1 and Tab. 2:
>
> |Method|Bit|FVD($\downarrow$)|VQA-Aesthetic($\uparrow$)|
> |-|-|-|-|
> |FP|16|101.9|66.91|
> |ViDiT-Q|W4A6|132.6|54.66|
> |**Ours**|W4A6|**103.6**|**67.05**|
> |ViDiT-Q|W3A6|251.8|39.82|
> |**Ours**|W3A6|**191.1**|**53.53**|
>
> Compared to baseline method ViDiT-Q, our method achieved lower FVD, demonstrating the consistent superiority of our method on both high and low level metrics.
>
> > Q4: Inference speed comparison.
>
> We have reported the inference efficiency of W4A8 model compared with FP model in Appendix Tab. 7. We also display the data here.
> |Method|Memory Cost($\downarrow$)|Latency Cost($\downarrow$)|VQA-Aesthetic($\uparrow$)|VQA-Technical($\uparrow$)|
> |-|-|-|-|-|
> |FP|10.9G|51.3s|66.91|53.49|
> |**Ours**|**4.5G (2.4$\times$)**|**38.0s (1.35$\times$)**|**71.32**|**55.56**|
>
> Compared to FP model, our method achieves complete lossless performance while bringing 2.4$\times$ reduction in memory cost and 1.35$\times$ inference acceleration. In the revised version, we will modify our layout to make this more prominent.
>
> > Q5: Motivation for temporal distillation.
>
> We quantitatively compare our distillation method with temporal differences:
>
> |Method|VQA-Aesthetic($\uparrow$)|VQA-Technical($\uparrow$)|
> |-|-|-|
> |No distillation|45.67|38.42|
> |Temporal differences|46.15|53.81|
> |**Ours**|**54.92**|**61.59**|
>
> As we discussed in Sec. 3.3, our motivation is **to perceive the inter-frame information of the whole video in distillation process. We hope to consider the overall information of the video while optimizing single frame information.** But direct MSE calculates different frame information separately as shown in Eq. 12. Although temporal differences can improve performance to some extent, our method achieved better results. **Concatenating all frames together can directly model the information of different frames in the whole video, which actually includes temporal differences between frames.** Our method can directly model the relationship matrix between all frames instead of simply the gap between two frames, so it can capture the global optimization information of the video well and achieve better performance.

---

> > ### Comment · Reviewer_paDY · 2025-04-06
> >
> > Thanks for addressing my concerns and for raising my score. The newly added visual comparisons are clear and significantly enhance the contribution of paper. It would be great to include comparisons with additional models—especially visual ones—in the main paper, although this might require major revisions.

---

> > > ### Author Response · Authors · 2025-04-06
> > >
> > > Dear reviewer,
> > >
> > > We sincerely appreciate your time and constructive feedback throughout the review process. We are delighted to hear that our rebuttal has addressed your concerns and that you now recommend acceptance. Your insightful comments have significantly strengthened our paper, and we are grateful for your valuable contribution to improving our work. We will include comparisons with additional models during rebuttal in the revised version.
> > >
> > > Best wishes,
> > >
> > > All authors

---

### Decision · Program_Chairs · 2025-05-01

**Decision:**

Accept (poster)

**Comment:**

This paper introduces a quantization framework designed specific for video diffusion transformers. The proposed method, Q-VDiT, is designed to address two challenges, the loss of information during quantization and the misalignment between optimization objectives and the unique requirements of video generation. It introduces Token aware Quantization Estimator (TQE) and Temporal Maintenance Distillation (TMD) to solve them respectively. Experiments show superiority of the new method again baselines in visual quality and quantitative performance in both high and low bitwidth settings.

The reviewers appreciate its clear presentation and strong empirical performance. The rebuttal addressed multiple concerns in the initial reviews, including the lack of low level metrics and analysis, lack of the original videos and speed comparison, and provided additional experiments on larger video generative models. It also explained the remaining artefacts observed in the generated videos. Although the reviewers did not participate in the post-rebuttal discussion, most of their concerns appear to have been sufficiently addressed.